# Avoid and Rule: Selective Sociality Scale for Understanding Introverted Personality in a Digitally Socialized World

**DOI:** 10.3390/ejihpe15060114

**Published:** 2025-06-17

**Authors:** Waqar Husain, Achraf Ammar, Khaled Trabelsi, Aseel AlSaleh, Haitham Jahrami

**Affiliations:** 1Department of Humanities, COMSATS University Islamabad, Islamabad Campus, Park Road, Islamabad 45550, Pakistan; 2Department of Training and Movement Science, Institute of Sport Science, Johannes Gutenberg-University Mainz, 55099 Mainz, Germany; acammar@uni-mainz.de; 3Research Laboratory, Molecular Bases of Human Pathology, LR19ES13, Faculty of Medicine of Sfax, University of Sfax, Sfax 3000, Tunisia; 4Research Laboratory Education, Motricité, Sport et Santé, EM2S, LR19JS01, High Institute of Sport and Physical Education of Sfax, University of Sfax, Sfax 3000, Tunisia; trabelsikhaled@gmail.com; 5Department of Movement Sciences and Sports Training, School of Sport Science, The University of Jordan, Amman 11942, Jordan; 6Department of Family and Community Medicine, College of Medicine and Health Sciences, Arabian Gulf University, Manama P.O. Box 12, Bahrain; aseelffs@agu.edu.bh; 7Government Hospitals, Manama P.O. Box 26671, Bahrain; 8Department of Psychiatry, College of Medicine and Health Sciences, Arabian Gulf University, Manama P.O. Box 26671, Bahrain

**Keywords:** selective sociality, socialization, social compliance, mental health, psychosocial health

## Abstract

(1) Background: The rapid expansion of information technology and globalization has significantly transformed psychosocial health, with excessive internet use and the pursuit of social acceptance via social media amplifying the impact of social factors, leading to both positive and negative psychological outcomes. This paper introduces “selective sociality,” a new psychological construct focused on mindful social engagement and digital interaction to maintain psychosocial health in the digital era, supported by a valid psychometric instrument: the Selective Sociality Scale (SSS). (2) Methods: The current research was conducted in a series of eight consecutive phases involving 1737 participants (Mean age = 24 years, SD = 7.66; women = 53.3%). The validation of the SSS involved exploratory and confirmatory factor analyses along with convergent, divergent, and predictive validity. (3) Results: The SSS demonstrated excellent reliability (α = 0.838; ICC = 0.954). The model fit indices, such as CFI (0.962), TLI (0.952), RMSEA (0.059), and SRMR (0.043), showed strong validity. The scale exhibited strong convergent validity with the Efficient Social Intelligence Scale. Selective sociality predicted positive personality traits and mental health but inversely predicted depression, anxiety, and stress. (4) Conclusions: The SSS offers valuable insights for both clinical practice and research.

## 1. Introduction

The understanding of mental health has progressed from early philosophical perspectives to a contemporary, multidimensional model that encompasses cognitive, emotional, sexual, social, environmental, religious, moral, and spiritual dimensions. In the digital age, excessive engagement with digital platforms has contributed to an increase in anxiety, depression, and detrimental behaviors associated with social media. In response to these challenges, this paper introduces the concept of selective sociality, a novel psychological and psychosocial strategy designed to enhance psychosocial health through intentional social engagement and mindful digital interaction. This construct is underpinned by theoretical and philosophical reasoning, and the paper also presents a validated psychometric tool for assessing selective sociality.

### 1.1. Mental Health Revisited: Contemporary Approaches and the Psychosocial Turn

The concept of mental health has evolved significantly over time. Initially, classical thinkers such as Aristotle associated mental health with happiness. Subsequently, psychological paradigms expanded upon this notion: the psychodynamic perspective highlighted the pleasure principle and the balance between the id, ego, and superego; cognitive theorists emphasized the importance of healthy thought processes; and social learning theorists focused on the influence of socialization on psychological functioning. Traditional conceptualizations often framed mental health as the achievement of life’s purpose ([8]; [51]), the fulfillment of needs and desires ([74], [75]; [110]), the maintenance of congruence between goals and accomplishments ([32]; [78]), and the experience of subjective life satisfaction ([6]; [72]; [102]). Empirical research has identified numerous correlates and predictors of mental health, including physical health and exercise ([11]; [31]; [92]), environmental conditions such as access to clean water and a healthy living environment ([124]), and psychosocial factors like body image ([76]), marital status ([69]), sexual satisfaction ([123]), and self-esteem ([39]). Furthermore, intrapersonal strengths associated with mental health include social intelligence, resilience, optimism, and effective decision-making ([20]; [23]; [81]; [82]). The quality of social support, family relationships, financial stability, and job satisfaction also emerge as critical predictors ([7]; [53]; [86]; [115]; [125]).

In recent years, there has been a paradigmatic shift from disorder-focused definitions to person-centered conceptualizations of mental health, favoring well-being and social-environmental factors over mere symptom absence ([4]; [38]; [41]; [54]; [64]; [96]; [101]; [108]; [112]). Mental health is now viewed as a multidimensional and lifelong process involving personal growth, positive relationships, social adaptation, creative functioning, and community contribution ([9]; [16]; [26]; [57]; [59]; [116]; [120]; [121]). This broadened view often uses terms like wellness, happiness, satisfaction with life, and quality of life interchangeably with mental health ([16]; [55]; [116]). The most recent development is the construct of psychosocial health, which integrates seven interlinked dimensions—socio-environmental, emotional, sexual, cognitive, religious, moral, and spiritual—offering a holistic alternative to traditional notions of mental health ([41]; [43]).

### 1.2. Recent Transformations in the Psychosocial Landscape Driven by Information Technology

The psychosocial landscape of the modern world has been fundamentally reshaped by the rapid proliferation of information technology (IT) and globalization. With nearly half of the global population owning smartphones and internet access becoming increasingly ubiquitous ([17]; [63]), digital connectivity has become a cornerstone of daily life. One of the most significant transformations has occurred in the domain of mental health, as online socialization via social media has become widespread ([83]). On a positive note, IT has facilitated the global dissemination of mental health awareness and resources. However, the digital revolution has also introduced substantial psychosocial risks. The heightened emphasis on social validation through online platforms has intensified the impact of social factors on mental health. The pursuit of belonging through digital interactions has led to an overvaluation of social approval, often symbolized by likes and followers, prompting many individuals to conform uncritically to prevailing online norms. This trend is exacerbated by exposure to idealized portrayals of others’ lives, which fuels unrealistic expectations and harmful self-evaluation.

Furthermore, digital overuse has been linked to several adverse mental health outcomes, including anxiety, depression, poor sleep, loneliness, cyberbullying, low body image satisfaction, self-harm, and suicidal ideation ([98]; [118]). New digital-age disorders such as internet addiction, nomophobia (fear of being without one’s phone), fear of missing out, and ghosting have emerged as maladaptive consequences of constant digital engagement ([21]; [47]; [49], [50]; [48]; [58]; [87]; [89]). As digital technologies continue to expand their global influence, it is imperative to re-evaluate and enhance our conceptual understanding of their complex impacts on mental health. This re-evaluation will facilitate the development of informed and effective interventions that promote psychosocial health in the digital age.

### 1.3. Selective Sociality and Its Significance for Psychosocial Health

Selective sociality is a psychosocial skill that enables individuals to consciously approach or avoid certain people, situations, or content with the genuine intention of enhancing their psychosocial health. It emphasizes selective social engagement, mindful digital interaction, and introspective well-being. By carefully selecting social encounters and activities, individuals can prevent unnecessary stress, social fatigue, and emotional exhaustion, which are often associated with excessive or superficial interactions. Selective sociality fosters positive engagements while minimizing exposure to toxic influences, particularly through the deliberate reduction of digital engagement. By limiting time spent on social media and avoiding mindless scrolling or online conflicts, individuals can focus on meaningful real-life relationships and activities that contribute to their emotional well-being. Instead of engaging in unproductive internet use, selective sociality encourages purposeful online activities such as education, professional development, and maintaining close relationships. Additionally, selective sociality emphasizes the value of listening over speaking, facilitating deeper and more meaningful conversations. It advocates prioritizing quality over quantity in social interactions, enabling individuals to build a strong, supportive network while conserving emotional energy. Strengthening familial bonds is also integral to this approach, as it avoids social obligations that feel burdensome rather than fulfilling. In addition to social interactions, selective sociality highlights the importance of introspection and self-reflection. In a world that often prioritizes external validation, taking time to assess one’s emotional state and behavioral patterns is essential for maintaining inner peace and making conscious choices aligned with personal values and long-term goals. Ultimately, selective sociality is not merely about avoiding certain individuals, situations, or content; It represents a comprehensive approach to life that prioritizes psychosocial health, meaningful connections, and personal fulfillment. By embracing selective sociality, individuals can create a balanced and enriching social life.

Selective sociality should not be conflated with other seemingly related constructs. For example, ‘social selection’ is recognized as a subtype of natural selection within biological studies ([122]). This concept examines the fitness of individuals based on the behaviors of others, reflecting reactions shaped by past experiences that lead individuals to avoid certain social interactions. Additionally, the term ‘locus of control’ pertains to an individual’s perceived capacity to influence and manage the circumstances of their life ([3]; [52]; [90]; [117]). Locus of control is classified into internal and external orientations ([93]), which subsequently inform labels such as ‘extraversion’ or ‘introversion’. ‘Avoidance’ is another related concept, which can be interpreted negatively ([28]; [38]) or positively ([34]). Individuals typically seek positive stimuli from their environment while avoiding negative stimuli ([27]). ‘Attention bias,’ as indicated by its name, refers to a cognitive bias rooted in selective attention ([24]), where certain stimuli receive more focus while others are disregarded ([106]). For instance, individuals with anxiety are more likely to direct their attention towards potential threats.

Selective sociality further differentiates itself from pertinent psychological theories, such as self-determination theory and social comparison theory. Self-determination theory ([25]) asserts that human well-being is driven by three fundamental psychological needs: autonomy, competence, and relatedness. Individuals are motivated to seek environments that satisfy these needs, fostering intrinsic motivation and personal growth. While Selective Sociality emphasizes autonomy, it does not prioritize intrinsic motivation or need fulfillment. Instead, it focuses on the intentional selection or avoidance of social interactions, digital content, and environments based on their potential effects on psychosocial health. Social comparison theory ([29]) elucidates how individuals assess their abilities, attitudes, and self-worth through comparisons with others. This theory posits that individuals engage in upward comparisons (with those perceived as superior) to motivate self-improvement or downward comparisons (with those perceived as worse off) to enhance self-esteem. In contrast, selective sociality is not centered on comparison but rather on intentional selection. It involves making conscious choices regarding whom to engage with, what digital content to consume, and which environments to inhabit. Rather than seeking validation or self-evaluation through comparison, selective sociality fosters self-preservation, emotional resilience, and psychological well-being by promoting interactions that positively contribute to one’s mental state.

### 1.4. Objectives of the Current Research

As a newly conceptualized construct, selective sociality necessitated the development and validation of a psychological scale capable of assessing the extent to which individuals integrate and apply this skill in their daily lives. Accordingly, the present research was undertaken to develop and validate the Selective Sociality Scale (SSS). The formulation and validation of the SSS represent a significant advancement in psychological assessment, facilitating the empirical measurement of individuals’ ability to engage in selective sociality across interpersonal and digital contexts. Employing a rigorous series of eight consecutive phases, the SSS was systematically developed and subjected to comprehensive psychometric evaluation via standardized methodologies to establish its validity and reliability. The resulting instrument provides a scientifically robust measure of selective sociality, offering valuable insights into its implications for psychosocial health within an increasingly technology-mediated social landscape.

## 2. Materials and Methods

The SSS was developed and validated through a series of eight rigorous phases of the current study to ensure its psychometric robustness and to analyze the beneficial effects of selective sociality on psychosocial health (Figure 1). Phase 1 involved the development of the scale, followed by exploratory factor analysis (EFA) to identify its underlying structure. Phase 2 employed confirmatory factor analysis (CFA) to establish the factorial validity of the scale. Phase 3 examined the correlation between selective sociality and an external locus of control. Phase 4 investigated the convergent validity of the SSS by correlating it with social intelligence, confirming its alignment with a theoretically related construct. Phase 5 tested the predictive validity of the scale in relation to positive personality traits (emotionality, creativity, sensitivity, responsibility, outlook, leadership, sympathy, justice, mercy, religiosity, and spirituality), establishing its role in fostering desirable psychological characteristics. Extending the predictive scope, Phase 6 explored the association between the SSS and mental health, including dimensions such as socioenvironmental, emotional, cognitive, moral, spiritual, and sexual wellness, as well as overall psychological and mental well-being. Phase 7 further assessed its predictive validity for psychopathological conditions, involving emotional, sexual, religious, moral, spiritual, and professional problems, along with depression, anxiety, and stress, demonstrating its relevance in overcoming psychopathological conditions. Finally, Phase 8 evaluated the predictive validity of the SSS in relation to psychosocial life satisfaction, providing empirical support for its role in enhancing overall satisfaction with life. Collectively, these phases of the current study establish the SSS as a psychometrically sound instrument, offering a comprehensive measure of an individual’s capacity to engage in selective sociality and its implications for psychosocial health.

### 2.1. Development of the SSS

An initial item pool of 24 items was constructed for the SSS. The development of these items was guided by the theoretical framework of selective sociality, emphasizing selective social engagement, mindful digital interaction, and introspective well-being. Each item was carefully crafted to capture key dimensions of selective sociality, including the prioritization of meaningful social connections, the regulation of social exposure to minimize stress, and the conscious management of digital engagement. Items such as “I prefer to spend time with a small group of genuine friends rather than engaging in large social gatherings” and “I carefully select whom I interact with to ensure that my social experiences are positive” reflect the importance of intentional relationship building. Others, such as “I limit my use of social media to prevent feelings of anxiety and dissatisfaction” and “I avoid aimless scrolling on the internet, focusing instead on meaningful online activities,” highlight the role of digital selectivity in promoting psychological well-being. Additionally, self-reflection and introspective well-being are central themes, as seen in items such as “I find time for introspection and self-reflection to maintain my emotional well-being” and “I consciously choose to engage in activities that contribute to my inner peace and contentment”. The inclusion of these items ensured that the scale comprehensively measures the construct of selective sociality by capturing both its interpersonal and digital dimensions.

The items were presented to a panel of five expert psychologists to assess the face validity of the Selective Sociality Scale (SSS). These experts possessed extensive experience in psychosocial and psychometric studies. The researchers provided the panel with an overview of the concept of selective sociality and the objectives of the proposed scale. The panel confirmed the validity of the items in relation to the construct of selective sociality. Their consensus was quantitatively assessed using interrater reliability, which indicated substantial agreement among the raters (Cohen’s weighted kappa = 0.782; Fleiss’s kappa = 0.780; Krippendorff’s alpha = 0.781). During the exploratory factor analysis, eleven items were excluded due to not meeting the necessary validity thresholds, such as communalities below 0.4 or cross-loadings exceeding 0.2 ([79]). The finalized SSS consists of 13 items and three distinct subscales, which were further validated through confirmatory factor analysis, along with the establishment of convergent, divergent, and predictive validity for the SSS.

### 2.2. Participants

The present research was conducted through a series of eight consecutive phases of the current study, each addressing a specific objective (Figure 1). A total of 1737 adults from Rawalpindi and Islamabad, Pakistan, participated in the study. The sample comprised 812 men (46.7%) and 925 women (53.3%). The participants’ ages ranged from 18 to 73 years, with a mean age of 24 years (SD = 7.66). In terms of marital status, 1405 (80.9%) were unmarried, while 332 (19.1%) were married. Educational qualifications varied from matriculation (10 years of formal schooling) to a doctoral degree, with the average level of education being graduation.

A convenience sampling technique was utilized to recruit participants for the study. Researchers approached individuals individually during visits to various academic institutions, government offices, and private organizations. Participation was entirely voluntary, and informed consent was obtained verbally from all participants prior to their involvement. Inclusion criteria required participants to be (a) at least 18 years old and (b) proficient in responding to questionnaires in English. The sample for this research adequately reflects the natural variability of social engagement styles within the population. While convenience sampling may be critiqued for potential selection bias, our recruitment process ensured an inclusive representation of both socially selective and socially active individuals, without systematically favoring either group. This approach allowed for a broad spectrum of individuals with varying degrees of social engagement to be included, as there were no predetermined criteria favoring one group over the other. Given that participation was voluntary, individuals retained the agency to choose whether to participate, further minimizing any systematic exclusion of specific social orientations.

### 2.3. Sample Size Calculations

The sample size for this research was determined based on established guidelines for EFA and CFA and to establish predictive validity. For EFA, a minimum of five participants per item is recommended ([15]; [114]). Given the initial pool of 24 items, the minimum sample size required for EFA was calculated to be 120 participants. Our data collection exceeded this requirement, with Phase 1 involving 149 participants. For CFA, the sample size requirement is generally greater, with the guideline recommending at least 10 participants per estimated parameter in the model ([60]). The CFA model in this study included 13 items and 3 factors, resulting in 29 parameters to be estimated (13 factor loadings, 13 error variances, and 3 covariances between factors). Thus, the minimum sample size required for CFA was calculated as 290 participants. This requirement was surpassed, with Phase 2 involving 387 participants, ensuring sufficient power to validate the factor structure. Predictive validity is typically assessed through regression or correlation analyses, which necessitate adequate statistical power to detect meaningful relationships between the scale scores and relevant outcome variables. A moderate effect size (f^2^ = 0.15) in a multiple regression analysis with three to five predictor variables requires a minimum sample of approximately 77 to 92 participants to achieve 80% power at an alpha level of 0.05 ([14]). However, to increase the robustness and generalizability of the findings, researchers often target a larger sample. The data collected in phases 3 to 8 adhered to this principle, with the minimum number of participants in any of these phases meeting or exceeding the required threshold.

### 2.4. Instruments

#### 2.4.1. Selective Sociality Scale (SSS)

The final SSS comprises 13 items (in English) and three distinct subscales: Selective Social Engagement (items 1 to 6), Mindful Digital Interaction (items 7 to 10), and Introspective Well-being (items 11 to 13). The response scale involves seven points, i.e., strongly disagree (scored 1), disagree (scored 2), slightly disagree (scored 3), not sure (scored 4), slightly agree (scored 5), agree (scored 6), and strongly agree (scored 7). The items are as follows: Item 1 = I avoid social situations that I believe will drain my emotional energy. Item 2 = I actively avoid situations or people who I know will cause me unnecessary stress. Item 3 = I avoid social situations that I know will lead to emotional fatigue or stress. Item 4 = I often disengage from social activities that do not contribute to my well-being. Item 5 = I choose to spend time with people who offer emotional support rather than with those who cause stress. Item 6 = I prefer to spend time with a small group of genuine friends rather than engaging in large social gatherings. Item 7 = I avoid aimless scrolling on the internet, focusing instead on meaningful online activities. Item 8 = I use the internet primarily for educational or professional purposes rather than for mindless browsing. Item 9 = I limit my use of social media to prevent feelings of anxiety and dissatisfaction. Item 10 = I restrict my screen time to focus more on meaningful real-life activities. Item 11 = I consciously choose to engage in activities that contribute to my inner peace and contentment. Item 12 = I value quiet time alone to recharge and reflect on my personal growth. Item 13 = I find time for introspection and self-reflection to maintain my emotional well-being. Sub-scales: Selective Social Engagement (items 1–6), Mindful Digital Interaction (items 7–10), Introspective Well-being (items 11–13).

#### 2.4.2. Rotter’s Locus of Control Scale

Selective sociality is closely aligned with an internal locus of control, as it involves conscious and intentional decisions about engaging or disengaging from social interactions and digital content that may affect well-being. Rotter’s locus of control scale ([95]) was utilized to correlate selective sociality with the external locus of control. It consists of 29 pairs of statements. Among these, six pairs serve as filler items designed to mask the scale’s intent, leaving 23 items that directly assess an individual’s locus of control, i.e., either internal or external. The respondents are required to choose the statement from each pair that they most agree with. The responses are scored to yield an overall locus of control score, with higher scores indicating a more external locus of control and lower scores indicating a more internal locus of control. The dichotomous nature of the scale allows for a clear classification of individuals along the internal–external continuum. The developers claimed the scale to be reliable and valid ([95]). The scale also exhibited satisfactory reliability in the current research (Cronbach’s alpha = 0.612).

#### 2.4.3. Efficient Social Intelligence Scale

Selective sociality builds upon the foundation of social intelligence by emphasizing intentional and mindful engagement in social and digital interactions that support psychosocial well-being. While social intelligence involves understanding and managing social dynamics, selective sociality applies this skill to selectively foster meaningful connections and set healthy social boundaries. The efficient social intelligence scale ([46]) was used to measure the convergent validity of the SSS. The scale comprises nine items and four subscales (knowledge, efficacy, relationships, and autonomy). It uses a seven-point Likert scale for responses (strongly disagree to strongly agree). The developer of the scale claimed it to be highly reliable (Cronbach’s alpha = 0.830 for the scale and ranging from 0.824 to 0.882 for the subscales; item–scale and item–total correlations were reported to be highly significant, *p* < 0.001) and valid (CFI = 0.990; TLI = 0.983; RMSEA = 0.045). The scale also exhibited satisfactory reliability in the current research (Cronbach’s alpha = 0.665).

#### 2.4.4. Personality and Character Scale

Selective sociality fosters positive personality traits by promoting self-awareness, emotional intelligence, and autonomy in social decision-making. By consciously engaging in meaningful social interactions and avoiding unnecessary or harmful engagements, individuals develop resilience, self-regulation, and psychological well-being. This intentional approach to socialization enhances positive personality traits. The Personality and Character Scale ([45]) was used to measure the predictive validity of the SSS for positive personality traits. The scale comprises 42 items and 11 subscales (emotionality, creativity, sensitivity, responsibility, outlook, leadership, sympathy, justice, mercy, religiosity, and spirituality). It uses a seven-point Likert scale for responses (strongly disagree to strongly agree). The developer of the scale claimed it to be highly reliable (Cronbach’s alpha = 0.933 for the scale and ranging from 0.853 to 0.924 for the subscales; item–scale and item–total correlations were reported to be highly significant, *p* < 0.001) and valid (CFI = 0.915; TLI = 0.904; RMSEA = 0.064). The scale also exhibited excellent reliability in the current research (Cronbach’s alpha = 0.920).

#### 2.4.5. Psychosocial Health Evaluator

Selective sociality enhances psychosocial health by allowing individuals to engage in meaningful social interactions while minimizing exposure to negative or draining social environments, thereby reducing stress and emotional exhaustion. The psychosocial health evaluator ([41]) was used to measure the predictive validity of the SSS for psychosocial health and its counterparts. The scale comprises 24 items and seven subscales (socioenvironmental wellness, emotional wellness, sexual wellness, cognitive wellness, religious wellness, moral wellness, and spiritual wellness). It uses a five-point Likert scale for responses (extremely incorrect to extremely correct). The developer of the scale claimed it to be highly reliable (Cronbach’s alpha = 0.851 for the scale and ranging from 0.420 to 0.957 for the subscales; item–scale and item–total correlations were reported to be highly significant, *p* < 0.001) and valid (CFI = 0.956; TLI = 0.948; RMSEA = 0.049). The scale also exhibited satisfactory reliability in the current research (Cronbach’s alpha = 0.679).

#### 2.4.6. Psychological Well-Being Scale

In addition to the psychosocial health evaluator, which measures mental health under the label of psychosocial health, the psychological well-being scale ([97]) was also used to measure the predictive validity of the SSS for psychological well-being to further establish the predictive validity of selective sociality for mental health. This widely used instrument is designed to assess multiple dimensions of psychological well-being, including autonomy, environmental mastery, personal growth, positive relationships with others, purpose in life, and self-acceptance. The scale comprises 18 items with a seven-point response scale ranging from strongly disagree to strongly agree. It has been extensively evaluated for its psychometric properties, demonstrating high internal consistency, with Cronbach’s alpha values typically ranging between 0.70 and 0.90 across different subscales. The scale also exhibited satisfactory reliability in the current research (Cronbach’s alpha = 0.615).

#### 2.4.7. Warwick–Edinburgh Mental Well-Being Scale

In addition to the psychosocial health evaluation scale and the psychological well-being scale, which measure mental health under the labels of psychosocial health and psychological well-being, the Warwick–Edinburgh mental well-being scale ([13]) was also used to measure the predictive validity of the SSS for mental well-being to further establish the predictive validity of selective sociality for mental health. The scale comprises 14 items with a five-point response sheet ranging from ‘none of the time’ to ‘all of the time.’ The developers of the scale reported it to be reliable (Cronbach’s alpha = 0.87). The scale also exhibited satisfactory reliability in the current research (Cronbach’s alpha = 0.717).

#### 2.4.8. Sukoon Psychosocial Illness Scale

To further demonstrate that selective sociality not only enhances psychosocial health but also aids in overcoming psychosocial challenges, the Sukoon Psychosocial Illness Scale ([43]) was used to measure the predictive validity of the SSS for psychosocial illness and its counterparts. The scale comprises 21 items and six subscales (emotional problems, sexual problems, social problems, professional problems, religious and moral problems, and spiritual problems). It uses a seven-point Likert scale for responses (very false to very true). The developer of the scale claimed it to be highly reliable (Cronbach’s alpha = 0.886 for the scale and ranging from 0.693 to 0.914 for the subscales; item–scale and item–total correlations were reported to be highly significant, *p* < 0.001) and valid (CFI = 0.942; TLI = 0.931; RMSEA = 0.055). The scale also exhibited good reliability in the current research (Cronbach’s alpha = 0.843).

#### 2.4.9. Depression, Anxiety, and Stress Scale

The depression, anxiety, and stress scale ([73]) was also used to measure the predictive validity of the SSS, specifically for depression, anxiety, and stress. The scale comprises 42 items and three subscales (depression, anxiety, and stress). It utilizes a four-point Likert scale for responses (did not apply to me at all—applied to me very much). The depression scale measures dysphoria, hopelessness, devaluation of life, self-deprecation, lack of interest/involvement, anhedonia, and inertia. The anxiety scale measures autonomic arousal, skeletal muscle effects, situational anxiety, and the subjective experience of anxious affect. The stress scale measures difficulty relaxing, nervous arousal, and being easily upset/agitated, irritable/overreactive, and impatient. It is a well-known scale and has been used in hundreds of studies worldwide ([12]; [18]; [42]). The scale also exhibited very good reliability in the current research (Cronbach’s alpha: Overall Scale = 0.893; Depression = 0.812; Anxiety = 0.722; Stress = 0.940).

#### 2.4.10. Psychosocial Life Satisfaction Scale

Life satisfaction is widely recognized as the ultimate indicator of an individual’s overall sense of fulfillment and well-being. The psychosocial life satisfaction scale ([40]) was used to measure the predictive validity of the SSS for psychosocial life satisfaction. The scale comprises five items and uses a seven-point Likert scale for responses (extremely unsatisfied to extremely satisfied). The developer of the scale claimed it to be highly reliable (Cronbach’s alpha = 0.872; item–total correlations were reported to be highly significant, *p* < 0.001) and valid (CFI = 0.991; TLI = 0.981; RMSEA = 0.066). The scale also exhibited good reliability in the current research (Cronbach’s alpha = 0.732).

### 2.5. Ethical Considerations

This research was conducted in accordance with established ethical guidelines (as outlined in the 1964 Helsinki Declaration and its subsequent amendments) to ensure the rights, dignity, and well-being of all participants. Ethical approval was obtained from the departmental review committee at COMSATS University (Code: CUI-ISB/HUM/ERC-CPA/2024-18) before data collection. Participation in the study was entirely voluntary, and informed consent was obtained from all participants prior to their inclusion. The participants were briefed about the purpose of the research, the confidentiality of their responses, and their right to withdraw at any stage without any consequences. Anonymity was maintained, and the participants were not asked for any personal information. No deceptive practices were used, and the participants did not face any potential risks or harm. The collected data were securely stored and used solely for research purposes, ensuring privacy and confidentiality.

### 2.6. Analysis

The collected data were recorded and analyzed using the Statistical Package for Social Sciences (SPSS-26) and R for statistical computing (R version 4.3.2). A rigorous data cleaning process was undertaken, which involved examining missing values, unengaged responses, outliers, linearity, homoscedasticity, multicollinearity, skewness, and kurtosis.

To assess the reliability and validity of the SSS, both EFA and CFA were conducted. In the EFA, principal axis factoring was employed, with the number of factors determined based on eigenvalues. A varimax rotation was applied to enhance interpretability. EFA was used to examine the factor structure, extraction values, Bartlett’s test of sphericity (BTS), Kaiser–Meyer–Olkin (KMO) measure of sampling adequacy, CFI, TLI, RMSEA, SRMR, and total variance explained.

For the CFA, the maximum likelihood extraction technique was utilized without rotation. Model fit was assessed via various fit indices, including the chi-square test, CFI, TLI, Bentler–Bonett non-normed fit index (NNFI), Bentler–Bonett normed fit index (NFI), Parsimony normed fit index (PNFI), Bollen’s relative fit index (RFI), Bollen’s incremental fit index (IFI), the relative noncentrality index (RNI), RMSEA, SRMR, the goodness-of-fit index (GFI), McDonald’s fit index (MFI), and the expected cross-validation index (ECVI). Additionally, the KMO test, Bartlett’s test of sphericity, the heterotrait–monotrait (HTMT) ratio, Cronbach’s alpha, and McDonald’s omega were computed to evaluate the scale’s reliability and construct validity.

To assess the correlation between variables, Pearson’s correlation coefficient was employed. Simple regression analysis was conducted to examine the predictive validity of the SSS. Furthermore, an independent samples *t*-test was performed to compare selective sociality levels between married and unmarried participants.

## 3. Results

### 3.1. Reliability

The SSS demonstrated good levels of reliability in both the EFA (Table 1; α = 0.838) and the CFA (Table 1; α = 0.817). The reliability of the subscales was also good in both the EFA (Table 1; α = 0.815, 0.808, and 0.714) and the CFA (Table 1; α = 0.808, 0.905, and 0.805). The item–total correlations (ranging from 0.295 to 0.684 with *p* < 0.001; mean = 0.586) and item–scale correlations (ranging from 0.576 to 0.854 with *p* < 0.001; mean = 0.763) of the SSS items, as measured via EFA, demonstrated a high degree of internal consistency (*p* < 0.001). The test–retest reliability of the SSS after a two-week interval with the same participants (n = 30) was excellent (intraclass correlation type = ICC3,1; point estimate = 0.954; lower 95% CI = 0.906; upper 95% CI = 0.978). The point estimates for the intraclass correlation for the three subscales of the SSS were also excellent (0.963, 0.924, and 0.954).

### 3.2. Exploratory Factor Analysis (EFA)

In the EFA (Table 2), principal axis factoring was employed, with the number of factors determined based on the eigenvalues. A varimax rotation was applied to enhance interpretability. The sampling adequacy was notable (Table 2; n = 149; KMO = 0.832 for the overall scale, which ranged from 0.776 to 0.881 for individual items). The adequacy of correlations between items was highly significant (Table 2; BTS: χ^2^ = 690.303, df = 78, *p* < 0.001). Eleven items were discarded during the exploratory factor analysis for not having the required thresholds for validity, such as communalities less than 0.4 or cross -loadings between factors above 0.2. The final SSS comprises 13 items and three distinct subscales: Selective Social Engagement (items 1 to 6); Mindful Digital Interaction (items 7 to 10); and Introspective Well-being (items 11 to 13). The factor loadings of these items ranged from 0.592 to 0.807 (Table 2). Several model fit indices, such as CFI (0.970), TLI (0.944), RMSEA (0.053), and SRMR (0.034), indicate strong validity.

### 3.3. Confirmatory Factor Analysis (CFA)

The CFA was conducted on 13 items to test a three-factor model. The maximum-likelihood extraction technique was utilized without rotation. The sampling adequacy was notable (Table 3; n = 387; KMO = 0.834 for the overall scale, which ranged from 0.760 to 0.899 for individual items). The adequacy of correlations between items was highly significant (Table 2; BTS: χ2 = 2254.899, df = 78, *p* < 0.001). The factor loadings were statistically significant (*p* < 0.001) and ranged from 0.436 to 0.862 (Table 3), indicating that the items were strongly related to the underlying factor. The average variances extracted for the three factors were 0.435, 0.705, and 0.591, respectively, demonstrating adequate convergence. The reliability was also good (coefficient ω = 0.886, coefficient α = 0.817). The CFA model demonstrated good fit according to several fit indices, such as CFI (0.962), TLI (0.952), NNFI (0.952), NFI (0.936), PNFI (0.744), RFI (0.919), IFI (0.962), RNI (0.962), RMSEA (0.059), SRMR (0.043), GFI (0.997), MFI (0.896), and ECVI (0.896).

### 3.4. Convergent Validity

Strong convergent validity was demonstrated by the scale’s highly substantial correlation with the Efficient Social Intelligence Scale (Table 4; r = 0.558, *p* < 0.001).

### 3.5. Predictive Validity

The predictive validity of the SSS (Table 5) was established through its strong predictive values for emotionality (β = 0.319; *p* < 0.001), creativity (β = 0.391; *p* < 0.001), sensitivity (β = 0.231; *p* < 0.001), responsibility (β = 0.466; *p* < 0.001), outlook (β = 0.409; *p* < 0.001), leadership (β = 0.244; *p* < 0.001), sympathy (β = 0.191; *p* < 0.01), justice (β = 0.165; *p* < 0.01), religiosity (β = 0.353; *p* < 0.001), spirituality (β = 0.329; *p* < 0.001), psychosocial health (β = 0.995; *p* < 0.001), socioenvironmental wellness (β = 0.589; *p* < 0.001), religious wellness (β = 0.413; *p* < 0.001), emotional wellness (β = 0.614; *p* < 0.001), cognitive wellness (β = 0.399; *p* < 0.001), moral wellness (β = 0.378; *p* < 0.001), spiritual wellness (β = 0.559; *p* < 0.001), sexual wellness (β = 0.663; *p* < 0.001), psychological well-being (β = 0.246; *p* < 0.001), mental well-being (β = 0.419; *p* < 0.001), psychosocial illness (β = −0.485; *p* < 0.001), emotional problems (β = −0.278; *p* < 0.001), sexual problems (β = −0.305; *p* < 0.001), religious and moral problems (β = −0.365; *p* < 0.001), social problems (β = −0.277; *p* < 0.001), spiritual problems (β = −0.125; *p* < 0.05), professional problems (β = −0.344; *p* < 0.001), depression (β = −0.467; *p* < 0.001), anxiety (β = −0.21; *p* < 0.001), stress (β = −0.223; *p* < 0.001), and psychosocial life satisfaction (β = 0.373; *p* < 0.001).

### 3.6. Selective Sociality, Locus of Control, Gender, Age, Education, and Marital Status

In addition to developing and validating the SSS, the current research also provides some interesting findings on selective sociality in relation to locus of control, gender, age, education, and marital status. A significant inverse correlation was found between selective sociality and external locus of control (Table 4; r = −0.781, *p* < 0.001). There were no significant differences in selective sociality across genders. However, married participants (n = 332) had significantly higher levels of selective sociality (M = 68.434, SD = 11.023, % = 75.202 vs. M = 65.747, SD = 10.312, % = 72.249; *p* < 0.001; Cohen’s d = 0.257) than unmarried participants (n = 1405). Age (Table 4; r = 0.181; *p* < 0.001) and education (Table 4; r = 0.097; *p* < 0.001) both had significant positive correlations with selective sociality.

## 4. Discussion

Selective sociality is a psychosocial skill that involves consciously choosing social interactions, digital engagement, and introspection to enhance psychosocial health. It helps individuals avoid unnecessary stress, social fatigue, and toxic influences—such as manipulative relationships, cyberbullying, excessive social comparison, online hostility, and emotionally draining interactions—by prioritizing meaningful relationships and limiting superficial interactions. This approach includes reducing mindless digital consumption, engaging in purposeful online activities, and fostering deep conversations by valuing listening over speaking. Selective sociality also emphasizes quality over quantity in social connections, strengthens familial bonds, and avoids burdensome social obligations. Beyond social interactions, it encourages self-reflection to maintain inner peace and align actions with personal values, ultimately fostering a balanced and fulfilling life.

To assess individuals’ ability to integrate selective sociality into daily life, the SSS was developed and validated in the present study through a series of eight consecutive phases. The scale underwent rigorous reliability and validity testing through both EFA and CFA. It has high internal consistency and test–retest reliability. Through statistical analyses, the final version of the SSS was refined to include 13 items divided into three factors. Selective social engagement is the first factor of the SSS. It involves avoiding social situations that cause stress and prioritizing emotionally supportive relationships. This factor emphasizes intentional choices in social interactions to preserve emotional energy and avoid stress. The items highlight a preference for meaningful and supportive relationships over draining or large social gatherings. Mindful digital interaction is the second factor of the SSS. It relates to limiting social media and internet use to meaningful activities. This factor reflects a conscientious approach to digital media and internet use, prioritizing purposeful activities and minimizing unproductive or anxiety-inducing engagement. Introspective well-being is the third factor of the SSS. It involves engaging in activities that promote inner peace and self-reflection. This factor captures the inclination toward activities and practices that foster inner peace, personal growth, and emotional well-being through introspection and solitude. Several model fit indices indicated excellent fit. The SSS was significantly correlated with the Efficient Social Intelligence Scale, confirming convergent validity. The SSS exhibited strong predictive power for various psychosocial constructs. The findings establish the SSS as a psychometrically robust measure of selective social engagement, mindful digital interaction, and introspective well-being. Its strong reliability, factor structure, and predictive validity underscore its utility in assessing social regulation processes and their impact on psychosocial health.

Beyond the development and validation of the SSS, the findings derived from this comprehensive series of eight phases of the current study offer novel theoretical and empirical contributions to the understanding of mental health in the contemporary digital landscape. The significant positive correlation between selective sociality and social intelligence indicates that individuals who engage in selective social interactions exhibit higher levels of social intelligence than those who do not. This finding suggests that the ability to discern and regulate social engagements can be linked to social intelligence, enabling individuals to navigate interpersonal dynamics more effectively and foster meaningful social connections. The results also substantiate that selective sociality serves as a significant positive predictor of several positive personality traits, such as emotionality, creativity, leadership, psychosocial health, spiritual wellness, and psychosocial life satisfaction. In the context of mental health, the present research examined selective sociality in relation to various dimensions of mental health, including mental well-being, psychological well-being, and psychosocial health. The findings revealed a significant positive correlation between selective sociality and all these facets of mental health, demonstrating that individuals who practice selective sociality exhibit better mental health outcomes than their counterparts. This underscores the role of intentional social engagement in fostering overall psychological resilience and well-being. Additionally, selective sociality was inversely correlated with psychosocial illness (emotional, sexual, religious, moral, spiritual, and professional problems), depression, anxiety, and stress. The positive correlations between selective sociality, age, and education further highlight the significance of selective sociality. These insights underscore the critical role of selective sociality in fostering psychological resilience and mitigating mental health vulnerabilities in an increasingly interconnected digital era.

The positive effects of selective sociality are consistent with several previous findings in the literature, which highlight the beneficial impact of intentional and purposeful social engagement on mental health and well-being. This study collectively underscores the importance of selective social interactions in promoting psychological resilience, reducing stress, and enhancing overall mental health outcomes. The concept of locus of control is significant to this discussion and refers to an individual’s perceived ability to influence and manage the circumstances surrounding their life ([3]; [52]; [90]; [117]). Locus of control is categorized into internal and external orientations ([93]). Those with an internal locus of control believe in their capacity to control their lives and shape events, whereas individuals with an external locus of control attribute control to external forces. People with an internal locus of control are likely to be more selective in their social interactions because they feel in control of their social environment. They tend to seek out relationships that align with their personal values, goals, and interests, as these are seen as extensions of their ability to influence outcomes. On the other hand, individuals with an external locus of control believe that external forces such as luck, fate, or the actions of others are primarily responsible for events in their lives. This belief can influence their approach to social interactions in distinct ways. They are more likely to seek out social interactions that provide reassurance or validation from others. Within the context of the current research, it is important to note that the literature has consistently associated the internal locus of control with several positive traits and behaviors. Individuals with an internal locus of control tend to engage in a more active pursuit of valued goals, exhibit an interest in seeking information ([67]; [104]), demonstrate alertness ([65]), engage in autonomous decision-making ([19]; [105]), and experience an enhanced sense of well-being ([67]; [68]; [111]). Conversely, studies have associated individuals with an external locus of control with higher levels of depression ([1]), anxiety, and a reduced ability to cope with stressful life experiences ([61]; [66]; [99]).

Extraversion and introversion are the two labels often associated with individuals exhibiting an external or internal locus of control. Extraversion is often associated with an external locus of control, whereas introversion is linked with an internal locus of control. These connections play a role in shaping how people approach and manage social interactions. Extroverts may thrive on external social engagements; they are typically more outgoing, sociable, and energized by interactions with others. They may be more inclined to perceive that external factors, such as social approval or relationships, significantly influence their success and well-being. This perception can lead to a greater emphasis on social engagements as a means of achieving desired outcomes. On the other hand, introverted individuals may achieve better mental health by limiting social interactions to those that are necessary or meaningful. They tend to be more reserved, introspective, and find energy in solitary activities or smaller, more intimate social settings. They are more likely to believe that their personal actions, thoughts, and decisions are the primary determinants of their life experiences. This belief aligns with a preference for self-directed activities and a tendency to limit social engagements to those that are meaningful or necessary. In today’s technologically advanced society, the undue significance placed on social interactions tends to lead to the social ostracization of introverts and those who deviate from societal norms. These norms inadvertently marginalize individuals who prefer solitude, introspection, or nonconformity. Mechanisms of ostracization may include social exclusion, reduced peer engagement, stereotyping (e.g., labeling introverts as antisocial), and algorithmic invisibility on digital platforms where visibility is tied to activity and conformity. In contrast to the commonly held misconception that introverts do not experience happiness ([37]), studies suggest that the quality of happiness for introverts differs from that of extroverts ([119]). Research indicates that introverts tend to achieve better mental health and well-being by selectively avoiding unnecessary social interactions and relationships ([37]; [119]). Additionally, distinguishing between shyness and introversion is essential, as they represent distinct constructs. While shy individuals fear social situations and negative judgments from others, introverts do not embrace such fears. Instead, introverts prefer unsociable activities and serene environments for their mental peace ([35]). Emotional stability, recognized as a robust predictor of happiness and well-being ([36]; [44]), is reported to be greater among introverts than among extroverts ([88]). Furthermore, solitude, which is strongly correlated with creativity and spirituality ([70]), is preferred by introverts over excessive social interactions ([35]; [37], [36]; [91]; [119]). Excessive sociality is viewed as oppressive ([71]), whereas solitude is linked with self-enhancement, spirituality, and improved mental health ([22]; [70]; [91]). Compared with extroverts, introverts exhibit a more thoughtful decision-making process, avoid impulsive decisions, and showcase superior decision-making skills ([56]). By maintaining a quiet and calm attitude in a world dominated by talkative and social extroverts, introverts experience enhanced well-being ([109]; [119]). Purposefully avoiding toxic individuals, situations, and unnecessary social interactions contributes to a more fulfilling life and promotes mental health ([100]).

Understanding the concepts of ‘avoidance’ and ‘escape’ from problems is also important for understanding ‘selective sociality’. Avoidance is construed both negatively ([28]; [38]) and positively ([34]). People tend to seek positive stimuli from their environment and avoid negative stimuli ([27]). Contentment is often found in ignorance, leading people to evade engaging in unpleasant tasks ([113]). Individuals motivated by avoidance, as opposed to those driven by approach, conserve energy by carefully selecting situations in which to exert cognitive effort ([94]). This process of selective exposure and avoidance, rooted in cognitive dissonance ([107]), helps individuals circumvent mental tension and attain a sense of peace ([103]). Notably, avoidance also involves unlearning, a pivotal factor in the enhancement of mental health ([33]). While learning involves a change in behavior, unlearning facilitates the discarding of outdated knowledge and behaviors. This paves the way for the assimilation of new information and the process of relearning ([5]; [10]). The management of cognitive bias in psychotherapy incorporates repeated approach–avoidance actions to diminish undesired behaviors ([27]). Furthermore, avoiding negative experiences and failures from the past is considered advantageous for professional success ([84]). The concept of ‘survival of the fittest’ also aligns with the idea of selected avoidance, particularly within the social context ([2]). Evolutionary psychology supports the perspective that species evolve to sustain behaviors and traits crucial for survival and reproduction ([62]). For example, rape avoidance is considered an evolutionary behavior that women develop to steer clear of situations where they might be vulnerable to rape ([85]). Social intelligence has emerged as a significant factor in predicting and interpreting human behavior within various social contexts ([30]). Social intelligence also involves the practice of selective avoidance ([80]). Individuals with greater social intelligence avoid unnecessary social interactions, possess persuasive abilities, and exhibit an awareness of the intentions, feelings, and vulnerabilities of others ([30]). The ability to selectively isolate from or interact with people reflects elevated social intelligence, which contributes to psychosocial well-being ([30]). Deviation from social norms is noted to be positively correlated with mental health, serving as a catalyst for changing existing social norms in one’s favor ([77]). Avoiding rigid cultural norms and engaging selectively in social tasks can promote mental peace and foster a sense of autonomy in shaping one’s own life ([77]; [103]; [107]). This strategic approach to social engagement features the interplay between selected avoidance, social intelligence, and mental well-being. This also indicates the fundamental role of selectivity as a guiding principle when navigating the extensive expanse of available information. The concept of selective sociality underscores the importance of intentional and purposeful social interactions for maintaining mental health and well-being. Introverts, in particular, benefit from limiting unnecessary social engagements, fostering a more fulfilling life that aligns with personal values and goals. Furthermore, the strategic management of social interactions, through selective exposure and avoidance, contributes significantly to mental peace, creativity, and overall well-being. This approach emphasizes the power of conscious social choices in navigating the complexities of modern, digitally mediated social environments.

The current research reveals no significant gender differences, likely due to evolving societal norms and more fluid gender roles, which have led both men and women to engage similarly in social behaviors such as seeking meaningful relationships and setting boundaries. Selective sociality is driven by universal needs for emotional well-being and personal growth, which transcend gender differences. However, marital status, age, and education have notable effects on selective sociality. Married individuals exhibit higher levels of selective sociality, prioritizing their spouses and setting stronger boundaries to protect their relationships, whereas unmarried individuals are more likely to engage in a diverse range of social interactions with greater spontaneity. As individuals age, they develop greater self-awareness and prioritize quality over quantity in their social interactions. Older adults, who have accumulated life experiences, tend to be more selective in their social engagements, focusing on emotionally fulfilling relationships. Similarly, higher education enhances cognitive and emotional intelligence, enabling individuals to navigate social interactions more effectively, make informed decisions, and foster meaningful connections by selectively disengaging from toxic relationships.

### 4.1. Suggestions to Improve Psychosocial Health Through Selective Sociality

The pursuit of social acceptance in the dynamic digital era has brought about a revolutionary change in the factors associated with mental health. One defining feature of modern life is active engagement in social events, whether in real life or online. The vast and interconnected world of social media is now a part of the social environment, which was once limited to physical communities. Likes and dislikes on social media sites have an outsized effect on people’s psychological well-being. These online affirmations or rejections have an impact that goes beyond the internet. This affects people’s sense of self-worth and their moods, for better or worse. An increasing number of people in the modern digital era are submitting to peer pressure and following fashion trends without considering the consequences of their choices. This unwise and irrational conformity to social norms highlights the relationship between group dynamics and psychological health. People who conform to social norms may lose their personal identity, lower their self-esteem, and develop several psychological problems, including stress, anxiety, and depression. To help people have a better relationship with the digitalized social landscape, it is essential to acknowledge the importance of having a detailed understanding and execution of selective sociality. Social interactions should be intentional and focused on fostering emotional support, belonging, and positive reinforcement. It is essential to nurture deep, meaningful relationships while minimizing superficial or draining interactions. Individuals should choose social events on the basis of their emotional energy and recognize when to withdraw from large gatherings or toxic environments to preserve psychosocial health. Avoiding unnecessary social obligations and stress-inducing relationships enhances well-being and emotional resilience. Cultivating an internal locus of control allows individuals to selectively influence their environment, leading to better emotional outcomes. Limiting digital consumption and engaging in online activities that align with personal values can reduce stress and anxiety. Additionally, integrating time for self-reflection through activities such as journaling or meditation promotes emotional stability and a balanced life.

With respect to psychosocial education, psychologists may create workshops or online courses to teach selective sociality skills, focusing on setting social boundaries, managing digital engagement, and fostering meaningful relationships. These sessions empower individuals to navigate social interactions mindfully, prioritize quality connections, and enhance emotional resilience. Additionally, introducing selective sociality as part of mental health curricula in schools and universities would help promote psychosocial health in younger populations. By equipping students with these essential skills, we can empower them to build authentic connections and safeguard their mental well-being in an increasingly digital world.

Therapists can assess clients’ ability to engage in selective social interactions via tools such as the SSS. Effective interventions include encouraging clients to prioritize supportive relationships, minimize emotionally draining interactions, and manage digital consumption. Setting boundaries around social media use, limiting exposure to negativity, and promoting digital detoxes can help reduce stress. Techniques such as guided journaling, mindfulness exercises, and solitary reflection can foster self-awareness and alignment with personal values, enhancing psychosocial health. Teaching clients to selectively disengage from toxic or unproductive interactions, set emotional boundaries, and develop social intelligence is key to improved psychosocial health. Additionally, therapy can challenge societal expectations that conflict with personal values, empowering clients to create authentic social environments that foster mental peace and autonomy.

### 4.2. Implications

The development and validation of the SSS have significant implications for both research and practice in psychology, particularly in the fields of social behavior, mental health, and digital engagement. First, the SSS provides a reliable tool for assessing individuals’ capacity for selective sociality, offering insights into how people navigate their social environments to maintain psychosocial health. By identifying individuals who struggle with social overstimulation, stress from unproductive relationships, or excessive digital consumption, practitioners can better tailor interventions to promote healthier social interactions and digital practices. From a therapeutic perspective, the SSS can be used to guide interventions that help individuals develop a more intentional and mindful approach to their social lives. The SSS offers a robust framework for exploring how selective sociality influences broader psychosocial outcomes, such as emotional resilience, stress management, and life satisfaction. It provides a standardized measure for examining the impact of social regulation processes across different populations, including those facing challenges related to social anxiety, digital addiction, or interpersonal conflict.

### 4.3. Limitations and Future Directions

Since the current study was performed within a specific collectivistic culture, its generalizability to other cultures, especially individualistic Western cultures, is limited. The difference in the values of individualistic and collectivistic cultures would have been addressed more appropriately if data had been collected cross-culturally. Future researchers are advised to assess selective sociality from a cross-cultural perspective to analyze item response patterns and social desirability. Additionally, applying item response theory (IRT) to evaluate the psychometric properties of the scale could offer deeper insights into how different items function across diverse populations and help identify any potential biases or cultural influences on responses.

## 5. Conclusions

Selective sociality is a concept that encourages the conscious selection of meaningful social interactions, the management of digital engagement, and the fostering of introspection to improve overall psychosocial health. By avoiding unnecessary stress, social fatigue, and toxic influences, individuals can focus on emotionally supportive relationships and reduce mindless digital consumption. This approach promotes quality over quantity in social connections, allowing people to prioritize deep, fulfilling relationships and activities that contribute to emotional well-being. Engaging in introspection also helps maintain inner peace and align actions with personal values. Studies have shown that individuals who practice selective sociality tend to exhibit better mental health outcomes, including higher emotional intelligence, resilience, and life satisfaction. This approach can be especially beneficial in today’s digital world, where intentional social engagement can help mitigate anxiety, stress, and depression. The Selective Sociality Scale, developed and validated in the current research, offers valuable insights for both clinical practice and research. The scale’s psychometric robustness and predictive power make it a valuable tool for assessing and promoting psychosocial health in diverse contexts.

## Figures and Tables

**Figure 1 ejihpe-15-00114-f001:**
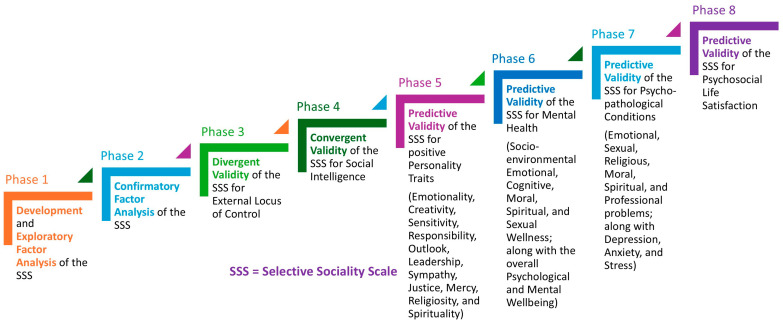
Scheme of the study.

**Table 1 ejihpe-15-00114-t001:** Descriptive statistics, reliability, and data accuracy (n = 1737).

Variable	Items	α	M	SD	%	Range		Skewness	Kurtosis
						Potential	Actual		
PHASE 1: n = 149; men = 72, 48.3%; women = 77, 51.7%; unmarried = 98, 65.8%; married = 51, 34.2%; age = 18–73 years, M = 27.27, SD = 10.839; education = matriculation to masters, average = graduation									
Selective Sociality Scale	13	0.838	67.463	11.469	74.135	13–91	32–91	−0.298	−0.003
Selective Social Engagement	6	0.815	34.342	5.850	81.767	6–42	14–42	−1.264	1.966
Mindful Digital Interaction	4	0.808	16.322	5.902	58.293	4–28	4–28	−0.118	−0.752
Introspective Well-being	3	0.714	16.799	3.093	79.994	3–21	6–21	−0.872	0.645
PHASE 2: n = 387; men = 192, 49.6%; women = 195, 49.4%; unmarried = 330, 85.3%; married = 57, 14.7%; age = 18–55 years, M = 24.21, SD = 5.788; education = matriculation to doctorate, average = graduation									
Selective Sociality Scale	13	0.817	68.615	8.729	75.401	13–91	41–89	−0.078	−0.382
Selective Social Engagement	6	0.808	35.328	4.134	84.115	6–42	21–42	−0.696	0.835
Mindful Digital Interaction	4	0.905	16.297	5.663	58.204	4–28	4–28	−0.079	−0.979
Introspective Well-being	3	0.805	16.990	2.474	80.903	3–21	11–21	−0.387	−0.525
PHASE 3: n = 217; men = 85, 39.2%; women = 132, 60.8%; unmarried = 180, 92.2%; married = 37, 17.1%; age = 18–54 years, M = 23.63, SD = 6.585; education = matriculation to doctorate, average = graduation									
Selective Sociality Scale	13	0.834	62.917	12.453	69.140	13–91	15–91	−0.380	1.149
Selective Social Engagement	6	0.824	32.051	6.780	76.311	6–42	6–42	−0.958	1.142
Mindful Digital Interaction	4	0.863	15.590	6.296	55.678	4–28	4–28	−0.010	−0.809
Introspective Well-being	3	0.791	15.277	3.912	72.745	3–21	3–21	−0.872	0.705
Rotter’s Locus of Control Scale	23	0.612	13.747	3.588	59.767	0–23	5–21	−0.409	−0.385
PHASE 4: n = 163; men = 89, 54.6%; women = 74, 45.4%; unmarried = 139, 85.3%; married = 24, 14.7%; age = 18–72 years, M = 23.08, SD = 6.676; education = matriculation to doctorate, average = graduation									
Selective Sociality Scale	13	0.834	65.963	11.633	72.487	13–91	23–90	−0.444	0.257
Selective Social Engagement	6	0.859	32.522	6.800	77.432	6–42	8–42	−1.007	0.723
Mindful Digital Interaction	4	0.849	17.166	5.835	61.306	4–28	4–28	−0.307	−0.628
Introspective Well-being	3	0.700	16.276	3.268	77.505	3–21	6–21	−0.883	0.591
Efficient Social Intelligence Scale	9	0.665	45.466	7.298	72.169	9–63	21–59	−0.314	0.032
PHASE 5: n = 264; men = 116, 43.9%; women = 148, 56.1%; unmarried = 180, 68.2%; married = 84, 31.8%; age = 18–64 years, M = 27, SD = 10.445; education = matriculation to doctorate, average = graduation									
Selective Sociality Scale	13	0.833	65.663	11.694	72.157	13–91	37–91	−0.070	−0.352
Selective Social Engagement	6	0.797	33.405	6.065	79.536	6–42	16–42	−0.791	0.057
Mindful Digital Interaction	4	0.860	16.318	6.073	58.279	4–28	4–28	0.085	−0.896
Introspective Well-being	3	0.685	15.939	3.385	75.902	3–21	6–21	−0.723	0.039
Personality & Character Scale	42	0.920	228.405	27.738	77.689	42–294	120–290	−0.459	0.582
PHASE 6: n = 163; men = 81, 49.7%; women = 82, 50.3%; unmarried = 180, 68.2%; married = 84, 31.8%; age = 18–64 years, M = 27, SD = 10.445; education = matriculation to doctorate, average = graduation									
Selective Sociality Scale	13	0.785	66.798	9.018	73.404	13–91	48–85	0.003	−0.808
Selective Social Engagement	6	0.754	34.440	4.489	82.000	6–42	21–42	−0.594	0.118
Mindful Digital Interaction	4	0.787	15.810	5.151	56.464	4–28	6–26	0.151	−0.804
Introspective Well-being	3	0.785	16.550	2.919	78.810	3–21	6–21	−1.059	1.506
Psychosocial Health Evaluator	24	0.679	87.620	6.684	73.016	24–120	73–105	0.056	−0.449
Psychological Well-being Scale	18	0.615	85.601	9.875	67.937	18–126	62–110	0.065	−0.411
Warwick–Edinburgh Mental Well-being Scale	14	0.717	41.607	5.991	59.439	14–70	24–56	−0.212	0.405
PHASE 7: n = 283; men = 120, 42.4%; women = 163, 57.6%; unmarried = 250, 88.3%; married = 33, 11.7%; age = 18–52 years, M = 22.93, SD = 5.461; education = matriculation to doctorate, average = graduation									
Selective Sociality Scale	13	0.735	65.537	9.022	72.019	13–91	49–91	0.324	−0.377
Selective Social Engagement	6	0.718	34.201	4.926	81.432	6–42	17–42	−0.644	0.308
Mindful Digital Interaction	4	0.817	15.028	5.454	53.673	4–28	4–28	0.285	−0.560
Introspective Well-being	3	0.767	16.307	2.820	77.654	3–21	7–21	−0.415	−0.329
Psychosocial Illness Scale	21	0.843	67.290	16.661	45.775	21–147	21–100	−0.328	−0.369
Depression, Anxiety, & Stress Scale	42	0.893	108.845	20.202	64.788	42–168	63–145	−0.059	−0.915
PHASE 8: n = 111; men = 57, 51.4%; women = 54, 48.6%; unmarried = 98, 88.3%; married = 13, 11.7%; age = 18–64 years, M = 22.59, SD = 5.686; education = matriculation to doctorate, average = graduation									
Selective Sociality Scale	13	0.757	65.883	9.511	72.399	13–91	44–90	0.160	−0.287
Selective Social Engagement	6	0.631	32.847	4.710	78.207	6–42	18–42	−0.748	0.746
Mindful Digital Interaction	4	0.823	16.640	5.569	59.427	4–28	4–28	−0.069	−0.696
Introspective Well-being	3	0.624	16.396	2.896	78.078	3–21	6–21	−0.736	0.650
Psychosocial Life Satisfaction	5	0.732	23.198	5.553	66.281	5–35	7–35	−0.537	0.477

Notes: n = number of participants; α = Cronbach’s Alpha; M = Mean; SD = Standard Deviation.

**Table 2 ejihpe-15-00114-t002:** Exploratory factor analysis and item–total and item–scale correlations (Study 1; n = 149).

Item No.	Item	Factor Structure			Ext.	Item–Total Correlation	Item–Scale Correlation		
		F1	F2	F3			F1	F2	F3
1	I avoid social situations that I believe will drain my emotional energy.	**0.750**	0.194	0.102	0.610	0.633 ***	**0.781 *****	0.294 ***	0.309 ***
2	I actively avoid situations or people who I know will cause me unnecessary stress.	**0.752**	0.218	0.127	0.629	0.646 ***	**0.77 *****	0.319 ***	0.331 ***
3	I avoid social situations that I know will lead to emotional fatigue or stress.	**0.744**	0.253	0.062	0.621	0.637 ***	**0.761 *****	0.338 ***	0.279 ***
4	I often disengage from social activities that do not contribute to my well-being.	**0.712**	0.230	0.116	0.574	0.638 ***	**0.751 *****	0.319 ***	0.336 ***
5	I choose to spend time with people who offer emotional support rather than with those who cause stress.	**0.667**	−0.011	0.311	0.542	0.558 ***	**0.699 *****	0.202 *	0.362 ***
6	I prefer to spend time with a small group of genuine friends rather than engaging in large social gatherings.	**0.592**	−0.274	0.058	0.429	0.295 ***	**0.576 *****	−0.044	0.088
7	I avoid aimless scrolling on the internet, focusing instead on meaningful online activities.	0.092	**0.807**	0.197	0.699	0.655 ***	0.249 **	**0.831 *****	0.374 ***
8	I use the internet primarily for educational or professional purposes rather than for mindless browsing.	0.157	**0.785**	0.217	0.687	0.684 ***	0.296 ***	**0.83 *****	0.391 ***
9	I limit my use of social media to prevent feelings of anxiety and dissatisfaction.	0.236	**0.747**	−0.010	0.613	0.622 ***	0.313 ***	**0.76 *****	0.263 **
10	I restrict my screen time to focus more on meaningful real-life activities.	0.002	**0.681**	0.318	0.565	0.592 ***	0.18 *	**0.767 *****	0.392 ***
11	I consciously choose to engage in activities that contribute to my inner peace and contentment.	0.268	0.165	**0.707**	0.600	0.578 ***	0.38 ***	0.346 ***	**0.766 *****
12	I value quiet time alone to recharge and reflect on my personal growth.	0.052	0.139	**0.784**	0.637	0.479 ***	0.223 **	0.302 ***	**0.777 *****
13	I find time for introspection and self-reflection to maintain my emotional well-being.	0.179	0.233	**0.763**	0.669	0.611 ***	0.335 ***	0.409 ***	**0.854 *****

Notes: F1 = Selective Social Engagement; F2 = Mindful Digital Interaction; F3 = Introspective Well-being; Ext. = Extraction/Communalities; Factor structure is Bold; * *p* < 0.05, ** *p* < 0.01, *** *p* < 0.001. Factoring method = Principal axis; Number of factors based on Eigenvalues; rotation = varimax. Bartlett’s Test of Sphericity: 690.303, df = 78, *p* < 0.001. Kaiser–Meyer–Olkin Measure of Sampling Adequacy: overall value = 0.832, values for individual items ranged from 0.776 to 0.881. Comparative Fit Index (CFI): 0.970; Tucker–Lewis Index (TLI): 0.944; Root mean square error of approximation (RMSEA): 0.053; Standardized root mean square residual (SRMR): 0.034; Total variance explained: 60.574. Sub-scales: Selective Social Engagement (items 1–6), Mindful Digital Interaction (items 7–10), Introspective Well-being (items 11–13). Response sheet: strongly disagree (scored 1), disagree (scored 2), slightly disagree (scored 3), not sure (scored 4), slightly agree (scored 5), agree (scored 6), strongly agree (scored 7).

**Table 3 ejihpe-15-00114-t003:** Confirmatory factor analysis (Study 2; n = 387).

Factor	Item	Factor Loadings				Residual Variances			
		Estimate	SE	z	*p*	Estimate	SE	z	*p*
F1	SSS1	0.701	0.051	14.749	<0.001	0.509	0.050	11.530	<0.001
	SSS2	0.837	0.044	18.720	<0.001	0.299	0.035	8.096	<0.001
	SSS3	0.719	0.042	15.231	<0.001	0.484	0.034	11.253	<0.001
	SSS4	0.572	0.050	11.345	<0.001	0.673	0.051	12.629	<0.001
	SSS5	0.589	0.049	11.778	<0.001	0.653	0.049	12.536	<0.001
	SSS6	0.436	0.047	8.305	<0.001	0.810	0.048	13.297	<0.001
F2	SSS7	0.840	0.069	19.775	<0.001	0.295	0.076	10.333	<0.001
	SSS8	0.862	0.067	20.563	<0.001	0.257	0.068	9.497	<0.001
	SSS9	0.815	0.070	18.877	<0.001	0.336	0.082	10.931	<0.001
	SSS10	0.844	0.066	19.876	<0.001	0.288	0.069	10.036	<0.001
F3	SSS11	0.699	0.047	14.143	<0.001	0.512	0.043	10.659	<0.001
	SSS12	0.812	0.046	17.309	<0.001	0.340	0.041	7.924	<0.001
	SSS13	0.785	0.047	16.659	<0.001	0.384	0.042	8.991	<0.001

Notes: F1 = Selective Social Engagement; F2 = Mindful Digital Interaction; F3 = Introspective Well-being. Extraction was performed using the Maximum-likelihood extraction technique with no rotation. Average variance extracted: F1 = 0.435; F2 = 0.705; F3 = 0.591. Chi-square test: Baseline model: χ^2^ = 2291.412, df = 78, Factor model: χ^2^ = 146.734, df = 62, *p* < 0.001; Additional Fit Measures: Comparative Fit Index (CFI): 0.962; Tucker–Lewis Index (TLI): 0.952; Bentler–Bonett Non-normed Fit Index (NNFI): 0.952; Bentler–Bonett Normed Fit Index (NFI): 0.936; Parsimony Normed Fit Index (PNFI): 0.744; Bollen’s Relative Fit Index (RFI): 0.919; Bollen’s Incremental Fit Index (IFI): 0.962; Relative Noncentrality Index (RNI): 0.962; Information Criteria: Log-likelihood: −6663.212, Number of free parameters: 42, Akaike (AIC): 13,410.425, Bayesian (BIC): 13,576.678, Sample-size adjusted Bayesian (SSABIC): 13,443.417; Root mean square error of approximation (RMSEA): 0.059; RMSEA 90% CI lower bound: 0.047; RMSEA 90% CI upper bound: 0.072; RMSEA *p*-value: 0.102; Standardized root mean square residual (SRMR): 0.043; Hoelter’s critical N (α = 0.05): 215.637; Hoelter’s critical N (α = 0.01): 240.483; Goodness of fit index (GFI): 0.997; McDonald fit index (MFI): 0.896; Expected cross-validation index (ECVI): 0.896; Kaiser–Meyer–Olkin (KMO) Test: Overall KMO: 0.834; KMO for individual indicators ranged from 0.760 to 0.899; Bartlett’s Test of Sphericity: χ^2^ = 2254.899, df = 78, *p* < 0.001; Heterotrait–monotrait ratio = 1; Reliability: Coefficient ω = 0.886, Coefficient α = 0.817.

**Table 4 ejihpe-15-00114-t004:** Correlations.

	Selective Sociality	Selective Social Engagement	Mindful Digital Interaction	Introspective Well-Being
External Locus of Control	−0.781 ***	−0.574 ***	−0.543 ***	−0.619 ***
Social Intelligence	0.558 ***	0.387 ***	0.431 ***	0.411 ***
Knowledge	0.333 ***	0.183 *	0.271 ***	0.320 ***
Efficacy	0.224 **	0.225 **	0.086	0.177 *
Relations	0.488 ***	0.324 ***	0.343 ***	0.450 ***
Autonomy	0.343 ***	0.223 **	0.362 ***	0.110
Emotionality	0.319 ***	0.036	0.381 ***	0.353 ***
Creativity	0.391 ***	0.109	0.389 ***	0.456 ***
Sensitivity	0.231 ***	0.111	0.197 **	0.245 ***
Responsibility	0.466 ***	0.311 ***	0.392 ***	0.348 ***
Outlook	0.409 ***	0.194 **	0.359 ***	0.423 ***
Leadership	0.244 ***	0.141 *	0.203 ***	0.224 ***
Sympathy	0.191 **	0.099	0.196 **	0.130 *
Justice	0.165 **	0.076	0.213 ***	0.050
Mercy	−0.003	−0.146 *	0.164 **	−0.044
Religiosity	0.353 ***	0.147 *	0.386 ***	0.264 ***
Spirituality	0.329 ***	0.112	0.322 ***	0.357 ***
Psychosocial Health	0.895 ***	0.671 ***	0.794 ***	0.640 ***
Socioenvironmental Wellness	0.589 ***	0.403 ***	0.465 ***	0.378 ***
Religious Wellness	0.413 ***	0.330 ***	0.330 ***	0.186 *
Emotional Wellness	0.614 ***	0.431 ***	0.512 ***	0.329 ***
Cognitive Wellness	0.399 ***	0.25 **	0.346 ***	0.237 **
Moral Wellness	0.378 ***	0.219 **	0.277 ***	0.340 ***
Spiritual Wellness	0.559 ***	0.377 ***	0.440 ***	0.371 ***
Sexual Wellness	0.663 ***	0.438 ***	0.517 ***	0.460 ***
Psychological Well-being	0.246 **	0.196 *	0.173 *	0.153
Mental Well-being	0.419 ***	0.245 **	0.363 ***	0.275 ***
Psychosocial Illness	−0.485 ***	−0.310 ***	−0.366 ***	−0.303 ***
Emotional Problems	−0.365 ***	−0.270 ***	−0.252 ***	−0.210 ***
Sexual Problems	−0.278 ***	−0.095	−0.278 ***	−0.186 **
Religious & Moral Problems	−0.305 ***	−0.240 ***	−0.201 ***	−0.167 **
Social Problems	−0.277 ***	−0.109	−0.238 ***	−0.234 ***
Spiritual Problems	−0.125 *	−0.109	−0.090	−0.035
Professional Problems	−0.344 ***	−0.249 ***	−0.233 ***	−0.216 ***
Depression	−0.467 ***	−0.350 ***	−0.262 ***	−0.376 ***
Anxiety	−0.210 ***	−0.074	−0.173 **	−0.208 ***
Stress	−0.223 ***	−0.192 **	−0.125 *	−0.137 *
Psychosocial Life Satisfaction	0.373 ***	0.163	0.366 ***	0.257 ***
Age	0.181 ***	0.086 ***	0.204 ***	0.078 ***
Education	0.097 ***	0.056 *	0.109 ***	0.026

Notes: * *p* < 0.05, ** *p* < 0.01, *** *p* < 0.001.

**Table 5 ejihpe-15-00114-t005:** Predictive validity of the SSS.

	R	R^2^	Adj. R^2^	df	F	B	SE B	β	t	*p*
Emotionality	0.319	0.102	0.098	262	29.614	0.257	0.047	0.319	5.442	<0.001
Creativity	0.391	0.153	0.149	262	47.194	0.177	0.026	0.391	6.870	<0.001
Sensitivity	0.231	0.053	0.050	262	14.738	0.055	0.014	0.231	3.839	<0.001
Responsibility	0.466	0.217	0.214	262	72.492	0.256	0.030	0.466	8.514	<0.001
Outlook	0.409	0.168	0.164	262	52.717	0.113	0.016	0.409	7.261	<0.001
Leadership	0.244	0.059	0.056	262	16.531	0.060	0.015	0.244	4.066	<0.001
Sympathy	0.191	0.036	0.033	262	9.875	0.045	0.014	0.191	3.142	< 0.01
Justice	0.165	0.027	0.023	262	7.319	0.034	0.013	0.165	2.705	< 0.01
Mercy	0.003	0.000	−0.004	262	0.003	−7.404	0.014	−0.003	−0.052	<0.001
Religiosity	0.353	0.125	0.121	262	37.358	0.163	0.027	0.353	6.112	<0.001
Spirituality	0.329	0.108	0.105	262	31.728	0.108	0.019	0.329	5.633	<0.001
Psychosocial Health	0.995	0.990	0.990	162	15,440.742	0.737	0.006	0.995	124.261	<0.001
Socioenvironmental Wellness	0.589	0.347	0.343	162	85.503	0.181	0.020	0.589	9.247	<0.001
Religious Wellness	0.413	0.170	0.170	162	33.043	0.064	0.011	0.413	5.748	<0.001
Emotional Wellness	0.614	0.377	0.373	162	97.240	0.107	0.011	0.614	9.861	<0.001
Cognitive Wellness	0.399	0.159	0.154	162	30.484	0.081	0.015	0.399	5.521	<0.001
Moral Wellness	0.378	0.143	0.137	162	26.784	0.074	0.014	0.378	5.175	<0.001
Spiritual Wellness	0.559	0.312	0.308	162	73.165	0.090	0.011	0.559	8.554	<0.001
Sexual Wellness	0.663	0.439	0.436	162	125.992	0.140	0.012	0.663	11.225	<0.001
Psychological Well-being	0.246	0.061	0.055	162	10.383	0.270	0.084	0.246	3.222	<0.001
Mental Well-being	0.419	0.175	0.170	162	34.233	0.278	0.048	0.419	5.851	<0.001
Psychosocial Illness	0.485	0.235	0.232	282	86.350	−0.895	0.096	−0.485	−9.292	<0.001
Emotional Problems	0.365	0.134	0.130	282	43.308	−0.308	0.047	−0.365	−6.581	<0.001
Sexual Problems	0.278	0.077	0.074	282	23.604	−0.118	0.024	−0.278	−4.858	<0.001
Religious & Moral Problems	0.305	0.093	0.090	282	28.786	−0.109	0.020	−0.305	−5.365	<0.001
Social Problems	0.277	0.077	0.073	282	23.306	−0.148	0.031	−0.277	−4.828	<0.001
Spiritual Problems	0.125	0.016	0.012	282	4.449	−0.045	0.021	−0.125	−2.109	<0.05
Professional Problems	0.344	0.118	0.115	282	37.765	−0.167	0.027	−0.344	−6.145	<0.001
Depression	0.467	0.218	0.216	282	78.545	−0.432	0.049	−0.467	−8.863	<0.001
Anxiety	0.210	0.044	0.041	282	12.971	−0.154	0.043	−0.210	−3.601	<0.001
Stress	0.223	0.050	0.046	282	14.735	−0.237	0.062	−0.223	−3.839	<0.001
Psychosocial Life Satisfaction	0.373	0.139	0.131	110	17.612	0.218	0.052	0.373	4.197	<0.001

Notes: *R* = Multiple correlation coefficient; *R*^2^ = Coefficient of determination; *Adj.* = Adjusted; *df* = Degrees of freedom; *F* = F statistic; *B* = Unstandardized regression coefficient; *SE* = Standard error; *β* = Standardized regression coefficient; *t* = t statistic; *p* = *p* value.

## Data Availability

The corresponding author can provide the data upon request.

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
