# Peer review of "Avoid and Rule: Selective Sociality Scale for Understanding Introverted Personality in a Digitally Socialized World"

_ejihpe, 2025, doi:10.3390/ejihpe15060114_

Round 1
Reviewer 1 Report (Previous Reviewer 3)
Comments and Suggestions for Authors
1) Participants should be described in a form of a table instead of describing this in table notes (lines 12679-1618) or in the text (lines 689-715). A table format will increase the readability.
2) Regarding Big Five Inventory, please indicate its reliability. In general, where are the descriptive statistics for this questionnaire? Regarding the DASS, please indicate its reliability for subscale scores as they were used in predictive validity analysis.
3) Lines 806-1018 should be moved to another place if the SSS is being introduced in the paper.
4) "The Big-5 Inventory [127] was used to establish the discriminant validity of the SSS.". I did not find any pieces of information regarding this analysis in the paper.
5) Descriptive for SSS subscales should be presented for each study. For instance, Studies 1 and 2 have data on descriptive statistics, but other studies did not (see Table 1).
6) The introduction should be more concise. For instance, a bunch of somewhat irrelevant citations does not look well in Section 1.1. Please clearly focus on the topic of this paper. Section 1.2 is irrelevant.
Author Response
Please see the report attached

Reviewer 2 Report (New Reviewer)
Comments and Suggestions for Authors
The manuscript highlights the novel contribution, has a robust validation scale, and takes into account demographic factors.
Some considerations in the discussion:
- While the introduction mentions avoiding "toxic influences," the discussion could benefit from a more explicit definition or examples of what constitutes a "toxic influence" in the context of selective sociality. Is it purely interpersonal, or does it extend to digital content or societal pressures?
- While the study establishes strong correlations and predictive power, the discussion occasionally verges on implying causality (e.g., "enabling individuals to navigate interpersonal dynamics more effectively"). It's important to maintain the distinction between correlation and causation, especially in correlational studies. Phrases like "is associated with," "contributes to," or "is consistent with" are generally safer.
- The statement "the undue significance placed on social interactions tends to lead to the social ostracization of introverts and those who deviate from societal norms" is a strong claim. While plausible, it could be further supported by more direct evidence or a deeper exploration of the societal pressures that lead to this "undue significance." What are the mechanisms of this ostracization?
Author Response
Please see the report attached

Round 2
Reviewer 1 Report (Previous Reviewer 3)
Comments and Suggestions for Authors
3) Lines 806-1018 should be moved to another place if the SSS is being introduced in the paper.
Authors’ response: We believe this may be a misunderstanding. Lines 806 to 1018 correspond to the entire Discussion section, which is correctly positioned well within the manuscript.
New comment: I am confused. Please the attached file which I received for review. In this file, lines 806-1018 are in the Methods, not in the discussion. In the revised version of paper, previous lines 806-1018 are placed in lines 320-338. This information should be removed from the current subsection 2.4.1, because these psychometric characteristics are described in the results section. Even lines 308-319 should be removed.
Examples of items in each subscale of each questionnaire should be included. Especially, it refers to the SSS.

Author Response
Second Response to Reviewer 1
I am confused. Please the attached file which I received for review. In this file, lines 806-1018 are in the Methods, not in the discussion. In the revised version of paper, previous lines 806-1018 are placed in lines 320-338. This information should be removed from the current subsection 2.4.1, because these psychometric characteristics are described in the results section. Even lines 308-319 should be removed.
Authors’ response: We agreed with the reviewer; thus, we deleted the psychometric characteristics that are described elsewhere in the results. Subsection 2.41 now solely focuses on a presentation of the Selective Sociality Scale (its items, subscales, and scoring) and reads as follows:
2.4.1 Selective Sociality Scale (SSS)
The final SSS comprises 13 items (in English) and three distinct subscales: Selective Social Engagement (items 1 to 6), Mindful Digital Interaction (items 7 to 10), and Introspective Wellbeing (items 11 to 13). The response scale involves seven points, i.e., strongly disagree (scored 1), disagree (scored 2), slightly disagree (scored 3), not sure (scored 4), slightly agree (scored 5), agree (scored 6), and strongly agree (scored 7). The items are as follows: Item 1 = I avoid social situations that I believe will drain my emotional energy. Item 2 = I actively avoid situations or people who I know will cause me unnecessary stress. Item 3 = I avoid social situations that I know will lead to emotional fatigue or stress. Item 4 = I often disengage from social activities that do not contribute to my well-being. Item 5 = I choose to spend time with people who offer emotional support rather than with those who cause stress. Item 6 = I prefer to spend time with a small group of genuine friends rather than engaging in large social gatherings. Item 7 = I avoid aimless scrolling on the internet, focusing instead on meaningful online activities. Item 8 = I use the internet primarily for educational or professional purposes rather than for mindless browsing. Item 9 = I limit my use of social media to prevent feelings of anxiety and dissatisfaction. Item 10 = I restrict my screen time to focus more on meaningful real-life activities. Item 11 = I consciously choose to engage in activities that contribute to my inner peace and contentment. Item 12 = I value quiet time alone to recharge and reflect on my personal growth. Item 13 = I find time for introspection and self-reflection to maintain my emotional well-being. Sub-scales: Selective Social Engagement (items 1-6), Mindful Digital Interaction (items 7-10), Introspective Wellbeing (items 11-13).

This manuscript is a resubmission of an earlier submission. The following is a list of the peer review reports and author responses from that submission.
Round 1
Reviewer 1 Report
Comments and Suggestions for Authors
The idea of selective sociality is a compelling one and I can definitely see potential theoretical as well as practical contribution to the literature and work on personality, social interactions and well-being. That being said the manuscript in its current form is not ready for publication and has numerous issues that require revision and re-writing before it can be shared with the world. Here are my comments to the authors:
1. The introduction need complete and deep re-writing: Most of it seems irrelevant and does not lay the theoretical ground for the new concept to be measured. I suggest the authors rewrite the introduction with the following structure in mind:
Start with the context - in one paragraph = what needs and lacunas in the current body of knowledge suggest the need for a new concept and a new measure?
Then define your new idea of selective sociality - define it and then differentiate it from similar concepts - for example: introversion, selective attention, attention bias, and others - in other words - convince the reader that this idea and concept is new, and different from existing concept within this field.
Then discuss ways of measuring what you defined - did you define it as a personality trait? a skill or ability? each has its own psychometric tradition of measurement - once you defined your concept you can review the ways of measuring it and similar concepts and then present the general path of the study - measure design and validation.
2. The sample section is confusing - I want to see details on demographics, and means of sampling your sample - so a reader can draw conclusions as to the relevance of this sample to his or her own target population or population of interest. Other descriptive statistics belong in the results chapter.
The instruments section MUST include a detailed description of the instrument's design, rationale, item generation process and preliminary content validity analyses at the very least. describe the structure of the measure and show how it corresponds to the theoretical definition you provide in the literature review. Also provide technical details of the other measures used (name, source and validity/ reliability information) for validation.
3. As suggested above - move the item generation and content validity analyses to the instrument description and provide more information. Add an appendix with the original list of items and the content validated list of items, preferably with the CVI grade for each.
4. Discriminant validity analyses in the results section - the section is incorrect. Discriminant validity does not mean inverse correlation (this is actually convergent) but the lack of association with competing concepts to show that it is different for example from introversion, or shyness.
5. Some of the tables in the results show 8 items and some 9. It needs to be clarified how many items were at the beginning of the validation and at which step items were dropped. Show results for reliability for the final version.
6. In the study limitations - the samples are taken mostly from Mediterranean , collectivistic cultures - the instrument could benefit from validation in western, individualistic culture settings. Language issues could modify and influence item response patterns, as well as social desirability.
I believe that once my concerns are properly addressed the manuscript can be seriously considered for publication.
Comments on the Quality of English LanguageA few minor point for editing. Can be done after the authors address the more essential concerns.
Author Response
Reviewer 1
The idea of selective sociality is a compelling one and I can definitely see potential theoretical as well as practical contribution to the literature and work on personality, social interactions and well-being. That being said the manuscript in its current form is not ready for publication and has numerous issues that require revision and re-writing before it can be shared with the world. Here are my comments to the authors:
Authors’ response: Thank you so much for your positive feedback and valuable input. We took all your comments seriously and carefully addressed all the identified corrections in the manuscript. We have thoroughly reviewed and revised the relevant sections to ensure accuracy and clarity. By incorporating your valuable feedback, we believe that the quality and effectiveness of our study have been significantly improved. We sincerely appreciate your recognition of the novelty and interest of our research. Your guidance and input have been invaluable in refining our work, and we are grateful for your support. Please note that the modifications done concerning your comments are colored in Yellow highlights in the revised manuscript.
- The introduction need complete and deep re-writing: Most of it seems irrelevant and does not lay the theoretical ground for the new concept to be measured. I suggest the authors rewrite the introduction with the following structure in mind:
Start with the context - in one paragraph = what needs and lacunas in the current body of knowledge suggest the need for a new concept and a new measure?
Then define your new idea of selective sociality - define it and then differentiate it from similar concepts - for example: introversion, selective attention, attention bias, and others - in other words - convince the reader that this idea and concept is new, and different from existing concept within this field.
Then discuss ways of measuring what you defined - did you define it as a personality trait? a skill or ability? each has its own psychometric tradition of measurement - once you defined your concept you can review the ways of measuring it and similar concepts and then present the general path of the study - measure design and validation.
Authors’ response: Thank you for your positive feedback. We rearranged the Introduction as you suggested by modifying the sequence, adding new text, and inserting sub-headings within the Introduction for clarity. We added new text and 60 new citations to rationalize the construct of selective sociality. Based on your valuable suggestions, the Introduction has been re-arranged and modified as under:
Construct of mental health
The construct of mental health has been viewed differently in history. Starting from Aristotle who viewed mental health as happiness [1], several theorists have been trying to elaborate the concept of mental health differently. The psychodynamic perspective emphasizes the role of ‘pleasure principle’ [2,3] and views mental health as the inner balance between id, ego, and super-ego [4]. The cognitive theorists focus on healthy cognition to attain mental health [5]. The social theorists emphasize on the role of social learning in mental health [6]. Conventionally, mental health has been viewed as the attainment of the purpose in life [7,8], the gratification of human needs [9,10], the satisfaction of human desires [11], a harmony between the desired and the achieved goals [12,13], and the subjective perception of a person [14–16]. Researchers have proposed several correlates and predictors of mental health such as adequate physical health [17,18], physical exercise [19], contact with water [20], a satisfactory body image [21], being married [22], sexual satisfaction [23], self-esteem [24], social intelligence, positive attitudes, responsible decision-making, self-awareness [25], extraversion in personality [26], psychological resilience [27], positive thinking [28], optimism for future [29], healthy family relations [30], social support [31,32], financial stability, job satisfaction [33–40], possession of a good living environment [41] and plenty of other psychosocial factors.
Contemporary approaches to mental health
The construct of mental health has seen a paradigm shift in the contemporary world. Person-centered definitions of mental health are more liked by psychologists than the diagnosis-based definitions [42,43]. In addition to the mere absence of mental disorders, several other constructs that project mental health have been incorporated to the construct of mental health. These factors mainly include the social and environmental factors related to mental health [44–50]. Mental health has been regarded as a multi-dimensional framework [51,52] and a capacity of a person to reach maximum growth, to maintain positive relationships, to adapt socially well, to work effectively and creatively, and to serve the community [53–57]. Mental health has been regarded as a lifelong process instead of an ultimate outcome [58]. Literature also suggests that the concepts of mental health, wellness, happiness, wellbeing, satisfaction with life, quality of life, have been used interchangeably [51,56,59]. The construct of psychosocial health is the most recent alternative to mental health [50]. It involves and combines seven interlinked dimensions of mental health under a single label of psychosocial health. These dimensions include socio-environmental, emotional, sexual, cognitive, religious, moral, and spiritual.
Recent changes in psychosocial environment through Information Technology
The contemporary world is deeply shaped by the widespread influence of information technology and globalization. Nearly half of the world's population owns a smartphone [60], and the Internet is becoming accessible to billions across the globe [61]. This rapid explosion of information technology (IT) on a global scale has significantly transformed various aspects of human life including mental health. The liberalization of information, the ease of global communication, and the empowerment of individuals through access to knowledge are all examples of the positive and constructive aspects of this integration of technology. On the positive side, the widespread accessibility of information through the internet has facilitated the dissemination of mental health awareness, resources, and support networks. Online platforms offer an excess of information on coping strategies, self-help techniques, and professional guidance, contributing to increased mental health literacy. Additionally, telemedicine and digital mental health interventions have emerged as viable alternatives for individuals seeking therapy or counseling, particularly in regions with limited access to traditional mental health services. However, the pervasive nature of IT also presents negative implications. Excessive use of social media platforms has been associated with psychological distress, anxiety, depression, feelings of isolation, compromised sleep quality, unfavorable indicators of mental health, contemplation of self-harm and suicide, instances of cyberbullying, dissatisfaction with body image, fear of missing out, and reduced life satisfaction [62,63]. The constant exposure to unrealistic and idealized representations of others' lives may contribute to unrealistic standards and social comparison, worsening mental health issues. Moreover, concerns about online privacy, cyberbullying, and information overload pose significant challenges. As the global integration of IT continues to evolve, a new understanding of its complex impact on mental health is imperative for developing effective interventions and promoting psychosocial well-being.
Social compliance
The adherence to "what others say" has consistently served as a persistent societal norm. Social compliance has always been regarded critically important [64], beneficial [65–67], and noble [68] to live a social life. It is assumed to be helpful in resolving conflicts within groups [69] and differentiating unique human behaviors within a societal context [70]. Studies reflect that the acquisition of social norms is a lifelong journey where individuals learn these norms through observation of others and their social environment [71–73]. The acquisition of social norms starts from the early childhood and is significantly influenced by electronic media [74]. Individuals who are more in social contact have more profound influences on shaping and reinforcing these norms [69,75,76]. Certain professions inherently involve extensive social contact, allowing individuals in these roles to significantly shape and reinforce societal norms. Politicians and public officials interact with the public through their policy decisions, speeches, and community engagements. Their role in shaping laws and regulations gives them considerable influence over societal norms. Teachers interact with students daily, influencing their values, beliefs, and knowledge. They play a key role in shaping the norms related to education, ethics, and social behavior. Individuals in the media industry have a wide reach and influence public opinion and societal norms through the content they produce and disseminate. They shape narratives around politics, culture, and social issues. Religious leaders often have significant influence over their congregations, shaping norms related to morality, ethics, and community behavior through sermons, counseling, and religious teachings. Those in the entertainment industry also have large audiences and influence norms related to fashion, lifestyle, and social issues through their performances, public appearances, and social media presence. People follow social norms to get social acceptance, to gain social rewards, and to be positively affiliated with the society [65,67,82,69,73,75,77–81]. They want to avoid social rejection [83] and punishment [67,69]. The impact of social compliance extends beyond mere adherence. It has a potential to change individuals' attitudes [69], personalities, communication [84], and behaviors [85].
Damages by excessive social engagement
The advent and widespread excess of information technology has worsened these concerns by expanding social circles and the numbers of those whose opinions weigh on individuals. Online socialization through various forms of social media has become an integral part of people's everyday lives around the world [86]. The quest for social acceptance through increased engagement on social media has elevated the importance of social aspects in shaping mental health. Social factors like social integration [87], participation in social events, being influenced by social environment, social acceptance, actualization, coherence, contribution, satisfaction, social comparison, public opinion, self-evaluation, inferiority and superiority complexes, quality of relationships, ghosting behaviors, and the fear of missing out in social media engagement have played a significant role in redefining mental health [88–93]. The opinions of others have gained significant importance, impacting mental health positively or negatively. The desire to enhance a sense of belongingness to social groups through online communication has made likes and dislikes excessively influential. Many individuals adhere to social norms blindly, often without understanding the rationale and effects of these trends. This unwise and illogical social compliance leads to various unhealthy consequences such as internet addiction (compulsive overuse of the internet that disrupts daily life), nomophobia (the fear of being without a mobile phone or losing signal), fear of missing out (anxiety about not being included in or missing out on exciting or important experiences others are having), and being ghosted (suddenly being ignored or cut off by someone without explanation, especially in social or digital communication) [92–99].
Divergence from the conventional understanding of mental health
The pursuit of social acceptance in the dynamic digital era has brought about a revolutionary change in the factors associated with mental health. One defining feature of modern life is the active engagement in social events, whether in real life or online. The vast and interconnected world of social media is now a part of the social environment, which was once limited to physical communities. Likes and dislikes on social media sites have an outsized effect on people's psychological wellbeing. These online affirmations or rejections have an impact that goes beyond the internet. This affects people's sense of self-worth and their moods, for better or worse. More and more people in the modern digital era are submitting to peer pressure and following fashion trends without giving any thought to the consequences of their choices. This unwise and irrational conformity to social norms highlights the relationship between group dynamics and psychological health. People conforming to social norms blindly may lose their personal identity, lower their self-esteem, and develop several psychological problems including stress, anxiety, and depression [100–103]. To help people have a better relationship with the digitalized social landscape, it is essential to acknowledge the importance of having a detailed understanding of these dynamics. This novel understanding is also mandatory for developing new psychotherapeutic interventions to cater to the mental health needs of people in today's technologically advanced society. Modern psychologists, therefore, have abandoned diagnosis-centric models in favor of person-focused ones that account for the complex interaction of variables impacting mental health [46]. The concept of mental health has evolved into something more complex, interconnected to many parts of a person's life, and is no longer defined by the absence of mental disorders alone [52]. Mental health is now recognized as an ever-changing, multi-faceted concept, marking a shift away from reductionist views [104]. Modern thought sees mental health care more as a journey than a destination. This view acknowledges that mental health is not an endpoint but rather an ongoing process affected by numerous internal and external factors [105].
Psychosocial health and selective sociality
The emergence of the 'psychosocial health model' signifies a transformative era in our understanding of mental well-being [50]. This innovative paradigm represents a profound departure from conventional perspectives [106], rooted in foundational research [107–113] and the convergence of religion, morality, spirituality, and psychology [114]. Within this paradigm, psychosocial health is defined as the comprehensive satisfaction of an individual's sexual, emotional, social, environmental, cognitive, religious, moral, and spiritual dimensions. This holistic perspective acknowledges the diversified human experiences that influence mental health, challenging the notion that social approval, recognition, and acceptance alone define one's mental wellbeing. Rather, it points out the importance of attending to the religious, moral, and spiritual aspects in fostering optimal mental health.
Central to the ‘psychosocial health model’ is the introduction of 'selective sociality'. Selective sociality is a psychosocial skill that enables individuals to approach or avoid certain individuals, situations, or content with the genuine intention of enhancing their psychosocial health. This psychosocial skill advocates for limiting electronic media usage, avoiding aimless internet activities, and minimizing unnecessary social engagements. Speaking less, being surrounded by real friends, strengthening emotional ties to family, limiting interactions with the public, using technology only for essential tasks, and sparing enough time for introspection and self-reflection are regarded as secrets to happiness and contentment, according to selective sociality. Selective sociality must not be confused with other apparently similar constructs. ‘Social selection’ is seen as a subtype of natural selection in the study of biology [115]. It addresses the fitness of individuals based on the behaviors of others. It is a reaction that depends on past experiences that motivates individuals to avoid others. ‘Locus of control’ refers to an individual's perceived ability to influence and manage the circumstances surrounding their life [116–119]. Locus of control is categorized into internal and external orientations [120] based on which people are labeled with ‘extraversion’ or ‘introversion’. ‘Avoidance’ is another similar concept that is taken both negatively [45,121] and positively [122]. People tend to seek positive stimuli from their environment and avoid the negative one [123]. ‘Attention bias’, as the name suggests, is a cognitive bias, based on selective attention [124], in which specific stimuli are given more attention while ignoring others [125]. For example, anxious individuals are prone to direct their attention more to the potential threats. The construct of ‘selective sociality’ presented here differs from these apparently similar constructs. Selective sociality is an important psychosocial skill that empowers individuals to navigate their social environments in such a selective way that leads to the enhancement of psychosocial health. This skill involves the ability to consciously choose when, how, and with whom to engage in social interactions. It enhances a person’s emotional and psychological resilience. By carefully selecting social encounters and activities, individuals can avoid the pitfalls of unnecessary stress, social fatigue, and emotional drain that often accompany excessive or meaningless social engagements. Selective sociality fosters positive interactions and minimizes exposure to negative or toxic influences. This includes the deliberate reduction of electronic media usage. By limiting time spent on social media and other forms of digital engagement, individuals can focus their energy on more meaningful, real-life relationships and activities that contribute to their mental and emotional well-being. Selective sociality encourages individuals to avoid aimless internet activities, such as mindless scrolling or engaging in online arguments, which can lead to feelings of anxiety, dissatisfaction, and social comparison. Instead, this skill promotes the use of the internet and technology in a purposeful and constructive manner, such as for educational purposes, professional development, or staying connected with close friends and family. Selective sociality emphasizes the importance of speaking less and listening more. This helps in fostering deeper and more meaningful conversations. By choosing to surround oneself with genuine friends who offer emotional support and encouragement, individuals can build a solid social network that nurtures their mental health. Strengthening emotional ties with family members is also a key component of selective sociality. Selective sociality advocates for limiting interactions with the public or engaging in social situations that do not contribute to one’s personal growth or happiness. This includes avoiding social obligations or events that feel more like burdens than opportunities for joy or connection. By prioritizing quality over quantity in social interactions, individuals can conserve their emotional energy and devote it to relationships and activities that truly matter. Selective sociality highlights the importance of carving out time for introspection and self-reflection. In a world that often prioritizes external achievements and social validation, taking time to look inward is essential for maintaining a sense of inner peace and contentment. Regular self-reflection enables individuals to assess their emotional state, recognize patterns of behavior that may be detrimental to their well-being, and make conscious choices that align with their values and long-term goals. Thus, selective sociality is not merely about unintentionally avoiding certain individuals, situations, or content. It is a comprehensive approach to life that prioritizes psychosocial health, meaningful connections, and personal fulfillment. Individuals can create a balanced and fulfilling social life through selective sociality.
The current study
As a newly proposed construct, there was a dire need for developing and validating a new psychological scale that could assess how well a person incorporates and utilizes selective sociality in life. The current research, therefore, was carried out to develop and validate the Selective Sociality Scale (SSS). The process involved a series of four consecutive studies. Employing standardized procedures, these studies aimed to establish the reliability and validity of the SSS, offering a valuable tool for measuring selective sociality in the context of psychosocial health.
- The sample section is confusing - I want to see details on demographics, and means of sampling your sample - so a reader can draw conclusions as to the relevance of this sample to his or her own target population or population of interest. Other descriptive statistics belong in the results chapter.
Authors’ response: The sample section has been modified in accordance with your valuable suggestions. Details on the demographics, including means, for each of the four studies have been added as follows:
The collective pool across the four studies comprised 688 respondents from Pakistan, Bahrain, and Tunisia. They were selected using a convenient sampling technique with only two inclusion criteria i.e. to be an adult (18 years of age or above) and to respond to questionnaires in English. English was not the first language of the participants. However, they were all educated enough to understand and respond to questionnaires in English. Responding to the questionnaires in English was the basic condition of their recruitment. The demographic composition of the total participants included both men (n=265; 38.5%) and women (n=423; 61.5%). Their ages ranged from 18 to 70 years, with an average age of 24 years. Study 1 involved 221 participants (men=62%; women=58%; age range=18-54 years, mean age=24 years). Study 2 involved 135 participants (men=31%; women=69%; age range=18-70 years, mean age=23 years). Study 3 involved 132 participants (men=46%; women=54%; age range=18-55 years, mean age=24 years). Study 4 involved 200 participants (men=35%; women=65%; age range=18-65 years, mean age=25 years). Data were collected during February and March 2024.
The instruments section MUST include a detailed description of the instrument's design, rationale, item generation process and preliminary content validity analyses at the very least. describe the structure of the measure and show how it corresponds to the theoretical definition you provide in the literature review. Also provide technical details of the other measures used (name, source and validity/ reliability information) for validation.
Authors’ response: We are thankful that you highlighted this. By putting headings for each scales used, we have provided the required information as suggested by you. The instruments are explained as follows:
Instruments
Selective Sociality Scale (SSS)
The Selective Sociality Scale (SSS) was developed and validated in the current series of four consecutive studies. The initial item-pool for SSS consisted of 24 items that were constructed based on a person’s interaction with the public, friends, and family. These items were observed by a panel of 5 PhD psychologists who determined the items to have appropriate face validity for the construct of selective sociality. After conducting exploratory and confirmatory factor analyses, the finalized version of the SSS includes 9 items (in English) and three distinct sub-scales, namely public (items 1 to 3), friends (items 4 to 6), and family (items 7 to 9). The response sheet involves seven points i.e. strongly disagree (scored 1), disagree (scored 2), slightly disagree (scored 3), not sure (scored 4), slightly agree (scored 5), agree (scored 6), strongly agree (scored 7). Across the four studies, the SSS demonstrated high test-retest reliability (Cronbach’s alpha= 0.862, 0.848, 0.890, 0.832). The reliability of the subscales was also good throughout the four studies (public: α=0.865, 0.853, 0.804, 0.921; friends: α=0.836, 0.867, 0.810, 0.879; family: α=0.838, 0.834, 0.802, 0.885). The item-total correlations (ranging from 0.447 to 0.787; mean= 0.687) and item-scale correlations (ranging from 0.670 to 0.908; mean= 0.828) of the SSS items demonstrated a high degree of internal consistency (p<0.01). Several model-fit indices showed strong validity (such as CFI=0.960; TLI=0.940) Strong convergent validity was demonstrated by the scale's highly substantial correlation with the Efficient Social Intelligence Scale (r=0.465, p<0.01). The discriminant validity was established through strong inverse correlations of SSS with the Big-5 Inventory (r= -0.709, p<0.01) and the external locus of control of the Rotter’s Locus of Control Scale (r= -0.742, p<0.01).
Efficient Social Intelligence Scale (ESIS)
The Efficient Social Intelligence Scale [126] was utilized to assess the convergent validity of the SSS. The ESIS comprises 9 items (in English) and includes four sub-scales i.e. knowledge, efficacy, relationships, and autonomy. The response sheet involves a 7-point Likert scale ranging from strongly disagree to strongly agree. The developer of the ESIS reported that the scale is highly reliable (α=830; average item-total correlation: r=.614; p<.001; average item-scale correlation: r=.913; p<.001) and valid (CFI=0.990; TLI=0.983; RMSEA=0.045).
Big Five Inventory
Big-5 Inventory [127] was used to establish the discriminant validity of the SSS. The Big Five Inventory is comprised of 44 items (short phrases) and measures the 5 personality traits i.e. extraversion vs introversion, agreeableness vs antagonism, conscientiousness vs lack of direction, neuroticism vs emotional stability, and openness vs closedness to experience. The inventory has been validated in many studies and has demonstrated adequate reliability and validity [128].
Rotter’s Locus of Control Scale
Rotter’s Locus of Control Scale [129] was utilized to measure the discriminant validity of the SSS. Rotter’s Locus of Control Scale consists of 29 pairs of statements. Among these, six pairs serve as filler items designed to mask the scale's intent, leaving 23 items that directly assess an individual's locus of control i.e. either internal or external. Respondents are required to choose the statement from each pair that they most agree with. The responses are scored to yield an overall locus of control score, with higher scores indicating a more external locus of control and lower scores indicating a more internal locus of control. The dichotomous nature of the scale allows for a clear classification of individuals along the internal-external continuum. The developers claimed the scale to be reliable and valid [129].
Demographic Information Questionnaire
A Demographic Information Questionnaire was also administered to gather details about participants' gender and age.
- As suggested above - move the item generation and content validity analyses to the instrument description and provide more information. Add an appendix with the original list of items and the content validated list of items, preferably with the CVI grade for each.
Authors’ response: The item generation and content validity analysis have been moved to the instrument description. More information on all the scales has been provided. An Appendix for the scale has also been added after tables.
- Discriminant validity analyses in the results section - the section is incorrect. Discriminant validity does not mean inverse correlation (this is actually convergent) but the lack of association with competing concepts to show that it is different for example from introversion, or shyness.
Authors’ response: We really appreciate your concerns. However, we have rechecked the terms used in our paper i.e. convergent and discriminant validity. Convergent validity, being a type of Criterion validity, is used to analyze the newly developed scale with another already validated scale based on a similar construct. The assumption here is that both the scales will be positively correlated. The Discriminant validity (also labeled as Divergent validity), on the other hand, is measured to establish that the newly developed scale is distinctive / inversely correlated with another already validated scale based on the conceptually distinct / opposite construct. In our case, we used social intelligence for convergent validity. The extraversion (sub-scale of the Big-Five Inventory) and the External Locus of Control (as measured by Rotter’s Locus of Control Scale) were taken as theoretically opposite constructs to selective sociality.
- Some of the tables in the results show 8 items and some 9. It needs to be clarified how many items were at the beginning of the validation and at which step items were dropped. Show results for reliability for the final version.
Authors’ response: Thank you for the detailed analysis. However, the items are 9 in all the tables as we rechecked those. The confusion might be a result of the numbering of the items which are presented according to their factors rather than being in numerical sequence.
- In the study limitations - the samples are taken mostly from Mediterranean , collectivistic cultures - the instrument could benefit from validation in western, individualistic culture settings. Language issues could modify and influence item response patterns, as well as social desirability.
Authors’ response: The suggestions regarding cultural issues, item response patterns, and social desirability have been incorporated into the Limitation as follows:
Since the current study was performed within a specific collectivistic culture, its generalizability to other cultures, especially the individualistic Western cultures, is also limited. The difference in the values of individualistic and collectivistic cultures would have been addressed more appropriately if data were collected cross-culturally. Future researchers are advised to assess selective sociality in a cross-cultural perspective to analyze item response patterns and social desirability.
I believe that once my concerns are properly addressed the manuscript can be seriously considered for publication.
Authors’ response: We are extremely grateful for your valuable suggestions. We assure you that we took all your suggestions very seriously and have incorporated them in the revised manuscript. We thank you for supporting and seriously considering our paper to be published.
Comments on the Quality of English Language
A few minor point for editing. Can be done after the authors address the more essential concerns.
Authors’ response: We have rechecked linguistic clarity, spelling, and grammar. We modified some text in this regard.

Reviewer 2 Report
Comments and Suggestions for Authors
Excellent paper! It adds valuable knowledge to the field and is of great interest.
In the Introduction, could we have a short review of other assessment instruments currently being used out in the field?
The Methods section could benefit from some descriptive demographic graphs, such as plot charts.
Could the Limitation of the study to be expanded to mention inevitable individual differences?
Author Response
Reviewer 2
Excellent paper! It adds valuable knowledge to the field and is of great interest.
Authors’ response: Thank you so much for your positive feedback and valuable input. We took all your comments seriously and carefully addressed all the identified corrections in the manuscript. We have thoroughly reviewed and revised the relevant sections to ensure accuracy and clarity. By incorporating your valuable feedback, we believe that the quality and effectiveness of our study have been significantly improved. We sincerely appreciate your recognition of the novelty and interest of our research. Your guidance and input have been invaluable in refining our work, and we are grateful for your support. Please note that the modifications done concerning your comments are colored in Yellow highlights in the revised manuscript.
In the Introduction, could we have a short review of other assessment instruments currently being used out in the field?
Authors’ response: Thank you for your concerns. Based on the suggestions provided by you and other reviewers, we have completely modified the Introduction. It contains sub-headings now, explaining a gradual rationale of conducting this study. Please note that ‘selective sociality’ is a new construct, presented for the very first time in our study. There are no earlier scales on this construct. However, we have discussed those constructs that appear similar to selective sociality such as social selection, locus of control, introversion, and avoidance. While adding new text, we have explained selective sociality further and discussed how it is different than the apparently similar constructs.
The Methods section could benefit from some descriptive demographic graphs, such as plot charts.
Authors’ response: We really appreciate your suggestion. However, please note that we have now added additional text to this paper by adding 60 more citations. We already have sufficient tables and graphs for this paper. Adding demographic graphs and plot charts will make the paper unnecessarily lengthier and may be out of the journal’s capacity for spacing.
Could the Limitation of the study to be expanded to mention inevitable individual differences?
Authors’ response: Thank you for this valuable suggestion. We have added new text to highlight individual differences in our Limitation section, as follows:
Moreover, the inevitable individual differences in each culture may also limit the generalizability of our scale.

Reviewer 3 Report
Comments and Suggestions for Authors
Abstract:
The title emphasizes "a digitally socialized world" ("Avoid and Rule: Selective Sociality Scale for understanding introverted personality in a digitally socialized world"). However, the abstract does not mention digital contexts or social media. If digital socialization is a key focus, it should be reflected in the abstract.
Line 24: Please decipher "SSS". Please check the use of abbreviations as there are a lot of inconsistencies.
Line 26: Why are only "women=61.5%" mentioned?
Line 27: The word "results" is repeated; please reconsider.
Introduction:
Lines 42-53: Language-wise, the paragraph is well written. However, it is lengthy and contains repeated ideas (e.g., regarding the impact of IT), which makes it redundant.
Some statements seem very generic (e.g., lines 43 – 46). I suggest reconsidering this paragraph for more clarity and precision. The authors could strengthen this paragraph (e.g., the positive and negative impact of IT) by linking studies to the arguments/ examples (there are no citations between lines 42 – 53 and 95 - 106).
Lines 71-72: "Individuals who are more in social contact have more profound influences on shaping and reinforcing these norms." Please provide specific examples.
Lines 92-93: Terms like "internet addiction, nomophobia, fear of missing out, and being ghosted by social contacts" are used. Could these be defined or explained briefly?
Lines 95-140: Please divide this very long paragraph into smaller ones.
Lines 104-105: "This foolish and irrational conformity to social norms..." is somewhat vague. Could references/ examples be provided? Moreover, the language (e.g., "foolish") is informal.
Lines 110-111: "Modern psychologists, therefore, have abandoned diagnosis-centric models in favour of person-focused ones." Can you elaborate on what these models entail and how they relate to selective sociality?
Lines 112-116: Please streamline; the ideas seem to be repeated.
Lines 128-130: I recommend providing a more detailed explanation of "selective sociality". The concept is introduced towards the end but lacks a detailed explanation of how it addresses the issues discussed earlier.
Materials and methods:
Lines 142-144: Please provide a brief overview of each study's purpose and key findings to give readers a clearer understanding of their contributions (e.g., "The initial study developed the 'Selective Sociality Scale (SSS)' and used exploratory factor analysis to ....").
Lines 148-149: Please define "selective sociality" more precisely to ensure consistency.
Procedure description:
(a) How were the participants recruited?
(b) When was the study conducted?
(c) Please indicate the residence of participants (with % of different categories) as well as participants' basic language.
(d) Participants were recruited in Pakistan, Bahrain, and Tunisia, but the language used in the questionnaires applied was English. Please describe how you ensured that participants understood language.
Statistical analyses:
(a) Please indicate the specific versions of the SPSS and AMOS.
(b) Not all conducted analyses were described in this section. Please make sure that this section describes all methodological and statistical aspects of the conducted analyses.
Discussion:
Lines 274-281: Please simplify the explanation of locus of control (Suggestion: "Locus of control, which is central to this discussion, refers to whether individuals feel they control their own lives (internal locus) or believe external forces dictate their circumstances (external locus).") How does the locus of control specifically impact one's ability to practice selective sociality?
Line 269: The three factors of the SSS are briefly mentioned but not described. Please reconsider.
Lines 287-295: The discussion on extraversion and introversion could be clarified. For now, this part is dense and could be difficult to follow (Suggestion: "Extraversion and introversion are linked to locus of control. Introverts, often marginalized in a society that values social interaction, may achieve better mental health by avoiding unnecessary social engagements".). Moreover, further insights on how extraversion and introversion influence the practice of selective sociality could be provided.
Conclusions:
I find the conclusions somewhat lengthy, especially their first part. I suggest reconsidering.
General comments:
While the article is well-written and maintains a smooth narrative flow, several elements could benefit from being shortened or simplified to enhance focus (examples were provided above). Additionally, incorporating more examples and references would strengthen the argumentation and provide more clarity and understanding to the readers. By addressing these aspects, the paper could achieve a more significant impact.
Author Response
Reviewer 3
Authors’ response: Thank you so much for your positive feedback and valuable input. We took all your comments seriously and carefully addressed all the identified corrections in the manuscript. We have thoroughly reviewed and revised the relevant sections to ensure accuracy and clarity. By incorporating your valuable feedback, we believe that the quality and effectiveness of our study have been significantly improved. We sincerely appreciate your recognition of the novelty and interest of our research. Your guidance and input have been invaluable in refining our work, and we are grateful for your support. Please note that the modifications done concerning your comments are colored in Yellow highlights in the revised manuscript.
Abstract:
The title emphasizes "a digitally socialized world" ("Avoid and Rule: Selective Sociality Scale for understanding introverted personality in a digitally socialized world"). However, the abstract does not mention digital contexts or social media. If digital socialization is a key focus, it should be reflected in the abstract.
Authors’ response: We thank you for correcting us. You are right. Digital socialization is not the key focus of our research. To avoid this confusion, we have changed the title of the study to be more specific with the objective of our study, i.e. the development and validation of selective sociality scale. The new title is as follows:
“Avoid and Rule: The development and validation of Selective Sociality Scale”
Line 24: Please decipher "SSS". Please check the use of abbreviations as there are a lot of inconsistencies.
Authors’ response: We thank you for highlighting this. The use of abbreviations has been rechecked. In Abstract, we have added the complete name of the instrument occurring for the first time, along with the abbreviation, to support further abbreviations in the abstract.
Line 26: Why are only "women=61.5%" mentioned?
Authors’ response: Based on your suggestion, we have added values for men as well. The text follows as:
The development and validation of the SSS involved four consecutive studies with a total of 688 adults (aged 18 to 70 with mean age = 24 years, men= 38.5%; women=61.5%).
Avoid and Rule: The development and validation of Selective Sociality Scale
Authors’ response: We have rechecked the repetition of the word “results” in line 27.
Introduction:
Lines 42-53: Language-wise, the paragraph is well written. However, it is lengthy and contains repeated ideas (e.g., regarding the impact of IT), which makes it redundant.
Some statements seem very generic (e.g., lines 43 – 46). I suggest reconsidering this paragraph for more clarity and precision. The authors could strengthen this paragraph (e.g., the positive and negative impact of IT) by linking studies to the arguments/ examples (there are no citations between lines 42 – 53 and 95 - 106).
Authors’ response: We appreciate you for such a detailed analysis and valuable suggestions. We have incorporated all your suggestions in true letter and spirit. Please note that the entire Introduction has been revised and restructured. Subheadings have been added for more clarity. A logical sequence is established to rationalize the study. The restructuring of Introduction and adding new text is as follows:
Construct of mental health
The construct of mental health has been viewed differently in history. Starting from Aristotle who viewed mental health as happiness [1], several theorists have been trying to elaborate the concept of mental health differently. The psychodynamic perspective emphasizes the role of ‘pleasure principle’ [2,3] and views mental health as the inner balance between id, ego, and super-ego [4]. The cognitive theorists focus on healthy cognition to attain mental health [5]. The social theorists emphasize on the role of social learning in mental health [6]. Conventionally, mental health has been viewed as the attainment of the purpose in life [7,8], the gratification of human needs [9,10], the satisfaction of human desires [11], a harmony between the desired and the achieved goals [12,13], and the subjective perception of a person [14–16]. Researchers have proposed several correlates and predictors of mental health such as adequate physical health [17,18], physical exercise [19], contact with water [20], a satisfactory body image [21], being married [22], sexual satisfaction [23], self-esteem [24], social intelligence, positive attitudes, responsible decision-making, self-awareness [25], extraversion in personality [26], psychological resilience [27], positive thinking [28], optimism for future [29], healthy family relations [30], social support [31,32], financial stability, job satisfaction [33–40], possession of a good living environment [41] and plenty of other psychosocial factors.
Contemporary approaches to mental health
The construct of mental health has seen a paradigm shift in the contemporary world. Person-centered definitions of mental health are more liked by psychologists than the diagnosis-based definitions [42,43]. In addition to the mere absence of mental disorders, several other constructs that project mental health have been incorporated to the construct of mental health. These factors mainly include the social and environmental factors related to mental health [44–50]. Mental health has been regarded as a multi-dimensional framework [51,52] and a capacity of a person to reach maximum growth, to maintain positive relationships, to adapt socially well, to work effectively and creatively, and to serve the community [53–57]. Mental health has been regarded as a lifelong process instead of an ultimate outcome [58]. Literature also suggests that the concepts of mental health, wellness, happiness, wellbeing, satisfaction with life, quality of life, have been used interchangeably [51,56,59]. The construct of psychosocial health is the most recent alternative to mental health [50]. It involves and combines seven interlinked dimensions of mental health under a single label of psychosocial health. These dimensions include socio-environmental, emotional, sexual, cognitive, religious, moral, and spiritual.
Recent changes in psychosocial environment through Information Technology
The contemporary world is deeply shaped by the widespread influence of information technology and globalization. Nearly half of the world's population owns a smartphone [60], and the Internet is becoming accessible to billions across the globe [61]. This rapid explosion of information technology (IT) on a global scale has significantly transformed various aspects of human life including mental health. The liberalization of information, the ease of global communication, and the empowerment of individuals through access to knowledge are all examples of the positive and constructive aspects of this integration of technology. On the positive side, the widespread accessibility of information through the internet has facilitated the dissemination of mental health awareness, resources, and support networks. Online platforms offer an excess of information on coping strategies, self-help techniques, and professional guidance, contributing to increased mental health literacy. Additionally, telemedicine and digital mental health interventions have emerged as viable alternatives for individuals seeking therapy or counseling, particularly in regions with limited access to traditional mental health services. However, the pervasive nature of IT also presents negative implications. Excessive use of social media platforms has been associated with psychological distress, anxiety, depression, feelings of isolation, compromised sleep quality, unfavorable indicators of mental health, contemplation of self-harm and suicide, instances of cyberbullying, dissatisfaction with body image, fear of missing out, and reduced life satisfaction [62,63]. The constant exposure to unrealistic and idealized representations of others' lives may contribute to unrealistic standards and social comparison, worsening mental health issues. Moreover, concerns about online privacy, cyberbullying, and information overload pose significant challenges. As the global integration of IT continues to evolve, a new understanding of its complex impact on mental health is imperative for developing effective interventions and promoting psychosocial well-being.
Social compliance
The adherence to "what others say" has consistently served as a persistent societal norm. Social compliance has always been regarded critically important [64], beneficial [65–67], and noble [68] to live a social life. It is assumed to be helpful in resolving conflicts within groups [69] and differentiating unique human behaviors within a societal context [70]. Studies reflect that the acquisition of social norms is a lifelong journey where individuals learn these norms through observation of others and their social environment [71–73]. The acquisition of social norms starts from the early childhood and is significantly influenced by electronic media [74]. Individuals who are more in social contact have more profound influences on shaping and reinforcing these norms [69,75,76]. Certain professions inherently involve extensive social contact, allowing individuals in these roles to significantly shape and reinforce societal norms. Politicians and public officials interact with the public through their policy decisions, speeches, and community engagements. Their role in shaping laws and regulations gives them considerable influence over societal norms. Teachers interact with students daily, influencing their values, beliefs, and knowledge. They play a key role in shaping the norms related to education, ethics, and social behavior. Individuals in the media industry have a wide reach and influence public opinion and societal norms through the content they produce and disseminate. They shape narratives around politics, culture, and social issues. Religious leaders often have significant influence over their congregations, shaping norms related to morality, ethics, and community behavior through sermons, counseling, and religious teachings. Those in the entertainment industry also have large audiences and influence norms related to fashion, lifestyle, and social issues through their performances, public appearances, and social media presence. People follow social norms to get social acceptance, to gain social rewards, and to be positively affiliated with the society [65,67,82,69,73,75,77–81]. They want to avoid social rejection [83] and punishment [67,69]. The impact of social compliance extends beyond mere adherence. It has a potential to change individuals' attitudes [69], personalities, communication [84], and behaviors [85].
Damages by excessive social engagement
The advent and widespread excess of information technology has worsened these concerns by expanding social circles and the numbers of those whose opinions weigh on individuals. Online socialization through various forms of social media has become an integral part of people's everyday lives around the world [86]. The quest for social acceptance through increased engagement on social media has elevated the importance of social aspects in shaping mental health. Social factors like social integration [87], participation in social events, being influenced by social environment, social acceptance, actualization, coherence, contribution, satisfaction, social comparison, public opinion, self-evaluation, inferiority and superiority complexes, quality of relationships, ghosting behaviors, and the fear of missing out in social media engagement have played a significant role in redefining mental health [88–93]. The opinions of others have gained significant importance, impacting mental health positively or negatively. The desire to enhance a sense of belongingness to social groups through online communication has made likes and dislikes excessively influential. Many individuals adhere to social norms blindly, often without understanding the rationale and effects of these trends. This unwise and illogical social compliance leads to various unhealthy consequences such as internet addiction (compulsive overuse of the internet that disrupts daily life), nomophobia (the fear of being without a mobile phone or losing signal), fear of missing out (anxiety about not being included in or missing out on exciting or important experiences others are having), and being ghosted (suddenly being ignored or cut off by someone without explanation, especially in social or digital communication) [92–99].
Divergence from the conventional understanding of mental health
The pursuit of social acceptance in the dynamic digital era has brought about a revolutionary change in the factors associated with mental health. One defining feature of modern life is the active engagement in social events, whether in real life or online. The vast and interconnected world of social media is now a part of the social environment, which was once limited to physical communities. Likes and dislikes on social media sites have an outsized effect on people's psychological wellbeing. These online affirmations or rejections have an impact that goes beyond the internet. This affects people's sense of self-worth and their moods, for better or worse. More and more people in the modern digital era are submitting to peer pressure and following fashion trends without giving any thought to the consequences of their choices. This unwise and irrational conformity to social norms highlights the relationship between group dynamics and psychological health. People conforming to social norms blindly may lose their personal identity, lower their self-esteem, and develop several psychological problems including stress, anxiety, and depression [100–103]. To help people have a better relationship with the digitalized social landscape, it is essential to acknowledge the importance of having a detailed understanding of these dynamics. This novel understanding is also mandatory for developing new psychotherapeutic interventions to cater to the mental health needs of people in today's technologically advanced society. Modern psychologists, therefore, have abandoned diagnosis-centric models in favor of person-focused ones that account for the complex interaction of variables impacting mental health [46]. The concept of mental health has evolved into something more complex, interconnected to many parts of a person's life, and is no longer defined by the absence of mental disorders alone [52]. Mental health is now recognized as an ever-changing, multi-faceted concept, marking a shift away from reductionist views [104]. Modern thought sees mental health care more as a journey than a destination. This view acknowledges that mental health is not an endpoint but rather an ongoing process affected by numerous internal and external factors [105].
Psychosocial health and selective sociality
The emergence of the 'psychosocial health model' signifies a transformative era in our understanding of mental well-being [50]. This innovative paradigm represents a profound departure from conventional perspectives [106], rooted in foundational research [107–113] and the convergence of religion, morality, spirituality, and psychology [114]. Within this paradigm, psychosocial health is defined as the comprehensive satisfaction of an individual's sexual, emotional, social, environmental, cognitive, religious, moral, and spiritual dimensions. This holistic perspective acknowledges the diversified human experiences that influence mental health, challenging the notion that social approval, recognition, and acceptance alone define one's mental wellbeing. Rather, it points out the importance of attending to the religious, moral, and spiritual aspects in fostering optimal mental health.
Central to the ‘psychosocial health model’ is the introduction of 'selective sociality'. Selective sociality is a psychosocial skill that enables individuals to approach or avoid certain individuals, situations, or content with the genuine intention of enhancing their psychosocial health. This psychosocial skill advocates for limiting electronic media usage, avoiding aimless internet activities, and minimizing unnecessary social engagements. Speaking less, being surrounded by real friends, strengthening emotional ties to family, limiting interactions with the public, using technology only for essential tasks, and sparing enough time for introspection and self-reflection are regarded as secrets to happiness and contentment, according to selective sociality. Selective sociality must not be confused with other apparently similar constructs. ‘Social selection’ is seen as a subtype of natural selection in the study of biology [115]. It addresses the fitness of individuals based on the behaviors of others. It is a reaction that depends on past experiences that motivates individuals to avoid others. ‘Locus of control’ refers to an individual's perceived ability to influence and manage the circumstances surrounding their life [116–119]. Locus of control is categorized into internal and external orientations [120] based on which people are labeled with ‘extraversion’ or ‘introversion’. ‘Avoidance’ is another similar concept that is taken both negatively [45,121] and positively [122]. People tend to seek positive stimuli from their environment and avoid the negative one [123]. ‘Attention bias’, as the name suggests, is a cognitive bias, based on selective attention [124], in which specific stimuli are given more attention while ignoring others [125]. For example, anxious individuals are prone to direct their attention more to the potential threats. The construct of ‘selective sociality’ presented here differs from these apparently similar constructs. Selective sociality is an important psychosocial skill that empowers individuals to navigate their social environments in such a selective way that leads to the enhancement of psychosocial health. This skill involves the ability to consciously choose when, how, and with whom to engage in social interactions. It enhances a person’s emotional and psychological resilience. By carefully selecting social encounters and activities, individuals can avoid the pitfalls of unnecessary stress, social fatigue, and emotional drain that often accompany excessive or meaningless social engagements. Selective sociality fosters positive interactions and minimizes exposure to negative or toxic influences. This includes the deliberate reduction of electronic media usage. By limiting time spent on social media and other forms of digital engagement, individuals can focus their energy on more meaningful, real-life relationships and activities that contribute to their mental and emotional well-being. Selective sociality encourages individuals to avoid aimless internet activities, such as mindless scrolling or engaging in online arguments, which can lead to feelings of anxiety, dissatisfaction, and social comparison. Instead, this skill promotes the use of the internet and technology in a purposeful and constructive manner, such as for educational purposes, professional development, or staying connected with close friends and family. Selective sociality emphasizes the importance of speaking less and listening more. This helps in fostering deeper and more meaningful conversations. By choosing to surround oneself with genuine friends who offer emotional support and encouragement, individuals can build a solid social network that nurtures their mental health. Strengthening emotional ties with family members is also a key component of selective sociality. Selective sociality advocates for limiting interactions with the public or engaging in social situations that do not contribute to one’s personal growth or happiness. This includes avoiding social obligations or events that feel more like burdens than opportunities for joy or connection. By prioritizing quality over quantity in social interactions, individuals can conserve their emotional energy and devote it to relationships and activities that truly matter. Selective sociality highlights the importance of carving out time for introspection and self-reflection. In a world that often prioritizes external achievements and social validation, taking time to look inward is essential for maintaining a sense of inner peace and contentment. Regular self-reflection enables individuals to assess their emotional state, recognize patterns of behavior that may be detrimental to their well-being, and make conscious choices that align with their values and long-term goals. Thus, selective sociality is not merely about unintentionally avoiding certain individuals, situations, or content. It is a comprehensive approach to life that prioritizes psychosocial health, meaningful connections, and personal fulfillment. Individuals can create a balanced and fulfilling social life through selective sociality.
The current study
As a newly proposed construct, there was a dire need for developing and validating a new psychological scale that could assess how well a person incorporates and utilizes selective sociality in life. The current research, therefore, was carried out to develop and validate the Selective Sociality Scale (SSS). The process involved a series of four consecutive studies. Employing standardized procedures, these studies aimed to establish the reliability and validity of the SSS, offering a valuable tool for measuring selective sociality in the context of psychosocial health.
Lines 71-72: "Individuals who are more in social contact have more profound influences on shaping and reinforcing these norms." Please provide specific examples.
Authors’ response: Thank you for giving us the opportunity to elaborate this. Following text has been added in this regard:
Certain professions inherently involve extensive social contact, allowing individuals in these roles to significantly shape and reinforce societal norms. Politicians and public officials interact with the public through their policy decisions, speeches, and community engagements. Their role in shaping laws and regulations gives them considerable influence over societal norms. Teachers interact with students daily, influencing their values, beliefs, and knowledge. They play a key role in shaping the norms related to education, ethics, and social behavior. Individuals in the media industry have a wide reach and influence public opinion and societal norms through the content they produce and disseminate. They shape narratives around politics, culture, and social issues. Religious leaders often have significant influence over their congregations, shaping norms related to morality, ethics, and community behavior through sermons, counseling, and religious teachings. Those in the entertainment industry also have large audiences and influence norms related to fashion, lifestyle, and social issues through their performances, public appearances, and social media presence.
Lines 92-93: Terms like "internet addiction, nomophobia, fear of missing out, and being ghosted by social contacts" are used. Could these be defined or explained briefly?
Authors’ response: Thank you for giving us the opportunity to elaborate this. Following text has been added in this regard:
This unwise and illogical social compliance leads to various unhealthy consequences such as internet addiction (compulsive overuse of the internet that disrupts daily life), nomophobia (the fear of being without a mobile phone or losing signal), fear of missing out (anxiety about not being included in or missing out on exciting or important experiences others are having), and being ghosted (suddenly being ignored or cut off by someone without explanation, especially in social or digital communication) [92–99].
Lines 95-140: Please divide this very long paragraph into smaller ones.
Authors’ response: We appreciate your suggestion. Please note that the entire Introduction has been revised and restructured. Subheadings have been added for more clarity.
Lines 104-105: "This foolish and irrational conformity to social norms..." is somewhat vague. Could references/ examples be provided? Moreover, the language (e.g., "foolish") is informal.
Authors’ response: Thank you for pointing this out. We modified the text by adding examples and references as follows:
This unwise and irrational conformity to social norms highlights the relationship between group dynamics and psychological health. People conforming to social norms blindly may lose their personal identity, lower their self-esteem, and develop several psychological problems including stress, anxiety, and depression [100–103].
Lines 110-111: "Modern psychologists, therefore, have abandoned diagnosis-centric models in favour of person-focused ones." Can you elaborate on what these models entail and how they relate to selective sociality?
Authors’ response: The explanation of models and their relevance to selective sociality has been clarified now by restructuring the paragraphs, adding headings to them, and rationalizing the sequence of the Introduction to finally conclude the significance of selective sociality. We hope that you would like this new structure, and this would surely address your concerns.
Lines 112-116: Please streamline; the ideas seem to be repeated.
Authors’ response: The entire Introduction has been streamlined by restructuring the paragraphs, adding headings to them, and rationalizing the sequence of the Introduction to finally conclude the significance of selective sociality. We hope that you would like this new structure, and this would surely address your concerns.
Lines 128-130: I recommend providing a more detailed explanation of "selective sociality". The concept is introduced towards the end but lacks a detailed explanation of how it addresses the issues discussed earlier.
Authors’ response: Thank you for encouraging us to explain selective sociality more. We have added new text in this regard as follows:
Psychosocial health and selective sociality
The emergence of the 'psychosocial health model' signifies a transformative era in our understanding of mental well-being [50]. This innovative paradigm represents a profound departure from conventional perspectives [106], rooted in foundational research [107–113] and the convergence of religion, morality, spirituality, and psychology [114]. Within this paradigm, psychosocial health is defined as the comprehensive satisfaction of an individual's sexual, emotional, social, environmental, cognitive, religious, moral, and spiritual dimensions. This holistic perspective acknowledges the diversified human experiences that influence mental health, challenging the notion that social approval, recognition, and acceptance alone define one's mental wellbeing. Rather, it points out the importance of attending to the religious, moral, and spiritual aspects in fostering optimal mental health.
Central to the ‘psychosocial health model’ is the introduction of 'selective sociality'. Selective sociality is a psychosocial skill that enables individuals to approach or avoid certain individuals, situations, or content with the genuine intention of enhancing their psychosocial health. This psychosocial skill advocates for limiting electronic media usage, avoiding aimless internet activities, and minimizing unnecessary social engagements. Speaking less, being surrounded by real friends, strengthening emotional ties to family, limiting interactions with the public, using technology only for essential tasks, and sparing enough time for introspection and self-reflection are regarded as secrets to happiness and contentment, according to selective sociality. Selective sociality must not be confused with other apparently similar constructs. ‘Social selection’ is seen as a subtype of natural selection in the study of biology [115]. It addresses the fitness of individuals based on the behaviors of others. It is a reaction that depends on past experiences that motivates individuals to avoid others. ‘Locus of control’ refers to an individual's perceived ability to influence and manage the circumstances surrounding their life [116–119]. Locus of control is categorized into internal and external orientations [120] based on which people are labeled with ‘extraversion’ or ‘introversion’. ‘Avoidance’ is another similar concept that is taken both negatively [45,121] and positively [122]. People tend to seek positive stimuli from their environment and avoid the negative one [123]. ‘Attention bias’, as the name suggests, is a cognitive bias, based on selective attention [124], in which specific stimuli are given more attention while ignoring others [125]. For example, anxious individuals are prone to direct their attention more to the potential threats. The construct of ‘selective sociality’ presented here differs from these apparently similar constructs. Selective sociality is an important psychosocial skill that empowers individuals to navigate their social environments in such a selective way that leads to the enhancement of psychosocial health. This skill involves the ability to consciously choose when, how, and with whom to engage in social interactions. It enhances a person’s emotional and psychological resilience. By carefully selecting social encounters and activities, individuals can avoid the pitfalls of unnecessary stress, social fatigue, and emotional drain that often accompany excessive or meaningless social engagements. Selective sociality fosters positive interactions and minimizes exposure to negative or toxic influences. This includes the deliberate reduction of electronic media usage. By limiting time spent on social media and other forms of digital engagement, individuals can focus their energy on more meaningful, real-life relationships and activities that contribute to their mental and emotional well-being. Selective sociality encourages individuals to avoid aimless internet activities, such as mindless scrolling or engaging in online arguments, which can lead to feelings of anxiety, dissatisfaction, and social comparison. Instead, this skill promotes the use of the internet and technology in a purposeful and constructive manner, such as for educational purposes, professional development, or staying connected with close friends and family. Selective sociality emphasizes the importance of speaking less and listening more. This helps in fostering deeper and more meaningful conversations. By choosing to surround oneself with genuine friends who offer emotional support and encouragement, individuals can build a solid social network that nurtures their mental health. Strengthening emotional ties with family members is also a key component of selective sociality. Selective sociality advocates for limiting interactions with the public or engaging in social situations that do not contribute to one’s personal growth or happiness. This includes avoiding social obligations or events that feel more like burdens than opportunities for joy or connection. By prioritizing quality over quantity in social interactions, individuals can conserve their emotional energy and devote it to relationships and activities that truly matter. Selective sociality highlights the importance of carving out time for introspection and self-reflection. In a world that often prioritizes external achievements and social validation, taking time to look inward is essential for maintaining a sense of inner peace and contentment. Regular self-reflection enables individuals to assess their emotional state, recognize patterns of behavior that may be detrimental to their well-being, and make conscious choices that align with their values and long-term goals. Thus, selective sociality is not merely about unintentionally avoiding certain individuals, situations, or content. It is a comprehensive approach to life that prioritizes psychosocial health, meaningful connections, and personal fulfillment. Individuals can create a balanced and fulfilling social life through selective sociality.
Materials and methods:
Lines 142-144: Please provide a brief overview of each study's purpose and key findings to give readers a clearer understanding of their contributions (e.g., "The initial study developed the 'Selective Sociality Scale (SSS)' and used exploratory factor analysis to ....").
Authors’ response: Based on your suggestions, we have modified this paragraph and have added the brief purpose and key findings of each study. The modified paragraph is as under:
This paper outlines the outcomes derived from a series of four consecutive studies. In the initial study, we developed the 'Selective Sociality Scale (SSS)' which comprised of 24 items in English. We conducted an exploratory factor analysis (EFA) of the scale which resulted in validating 9 items divided into 3 factors. Subsequently, the second study involved the confirmatory factor analysis (CFA) of the SSS. The CFA confirmed the factor structure, demonstrating good model fit (CFI: 0.960, RMSEA: 0.080). The SSS showed high internal consistency throughout the 4 studies (α= 0.862, 0.848, 0.890, 0.832). The third study examined the convergent validity of the SSS, establishing correlations between selective sociality and social intelligence (r=0.465, p<0.01). Moving forward, the fourth study focused on the discriminant validity of the SSS, establishing connections between selective sociality, extroversion personality trait (r=-0.709, p<0.01), and locus of control (r= -0.742; p<0.01). Selective sociality, as operationally defined during these studies, is a psychosocial skill to approach or avoid a person, situation, or content with a genuine intention to enhance psychosocial health.
Lines 148-149: Please define "selective sociality" more precisely to ensure consistency.
Authors’ response: Selective sociality has been precisely defined as follows:
Selective sociality, as operationally defined during these studies, is a psychosocial skill to approach or avoid a person, situation, or content with a genuine intention to enhance psychosocial health.
Please also note that we have modified the details about selective sociality in the Introduction as well.
Procedure description:
(a) How were the participants recruited?
(b) When was the study conducted?
(c) Please indicate the residence of participants (with % of different categories) as well as participants' basic language.
(d) Participants were recruited in Pakistan, Bahrain, and Tunisia, but the language used in the questionnaires applied was English. Please describe how you ensured that participants understood language.
Authors’ response: Thank you for your valuable additions. We have addressed these issues and the modified paragraph is as follows:
The collective pool across the four studies comprised 688 respondents from Pakistan, Bahrain, and Tunisia. They were selected using a convenient sampling technique with only two inclusion criteria i.e. to be an adult (18 years of age or above) and to respond to questionnaires in English. English was not the first language of the participants. However, they were all educated enough to understand and respond to questionnaires in English. Responding to the questionnaires in English was the basic condition of their recruitment. The demographic composition of the total participants included both men (n=265; 38.5%) and women (n=423; 61.5%). Their ages ranged from 18 to 70 years, with an average age of 24 years. Study 1 involved 221 participants (men=62%; women=58%; age range=18-54 years, mean age=24 years). Study 2 involved 135 participants (men=31%; women=69%; age range=18-70 years, mean age=23 years). Study 3 involved 132 participants (men=46%; women=54%; age range=18-55 years, mean age=24 years). Study 4 involved 200 participants (men=35%; women=65%; age range=18-65 years, mean age=25 years). Data were collected during February and March 2024.
Statistical analyses:
- Please indicate the specific versions of the SPSS and AMOS.
Authors’ response: Ther versions have been included as follows:
The collected data were recorded and analyzed utilizing the Statistical Package for Social Sciences (SPSS-26) and Analysis of Moment Structures (AMOS-20).
- Not all conducted analyses were described in this section. Please make sure that this section describes all methodological and statistical aspects of the conducted analyses.
Authors’ response: Thank you for your valuable concerns. We have rechecked for the inclusion of all the statistical procedures involved. The sentences are as follows:
A thorough data cleaning process was executed, involving the examination of missing values, unengaged responses, outliers, linearity, homoscedasticity, multicollinearity, skewness, and kurtosis. To assess the reliability and validity of the Selective Sociality Scale (SSS), both exploratory and confirmatory factor analyses were conducted. Additionally, Pearson Correlation Coefficient and descriptive statistics were employed to further illuminate patterns and relationships within the dataset.
Discussion:
Lines 274-281: Please simplify the explanation of locus of control (Suggestion: "Locus of control, which is central to this discussion, refers to whether individuals feel they control their own lives (internal locus) or believe external forces dictate their circumstances (external locus).") How does the locus of control specifically impact one's ability to practice selective sociality?
Authors’ response: Thank you for your valuable suggestions. We have added the following sentences to establish the link between locus of control and selective sociality:
People with an internal locus of control are likely to be more selective in their social interactions because they feel in control of their social environment. They tend to seek out relationships that align with their personal values, goals, and interests, as these are seen as extensions of their ability to influence outcomes. On the other hand, individuals with an external locus of control believe that external forces such as luck, fate, or the actions of others are primarily responsible for the events in their lives. This belief can influence their approach to social interactions in distinct ways. They are more likely to seek out social interactions that provide reassurance or validation from others.
Line 269: The three factors of the SSS are briefly mentioned but not described. Please reconsider.
Authors’ response: Thank you for your valuable suggestion. We have added the following sentences to elaborate more on public, friends, and family and their specific roles in being socially selective:
Social interactions with the public, friends, and family differ in their nature and scope. Public interactions are characterized by formality, limited emotional investment, and a focus on social roles and norms. Friendships offer informality, shared interests, and emotional support, with an emphasis on reciprocity and mutuality. Family interactions, on the other hand, involve deep emotional bonds, a sense of obligation, and complex intergenerational dynamics. Understanding these distinctions can provide valuable insights into how individuals navigate their social worlds and manage relationships across different contexts. The SSS, therefore, addresses these three distinctive social interactions to provide a more specialized assessment of selective sociality.
Lines 287-295: The discussion on extraversion and introversion could be clarified. For now, this part is dense and could be difficult to follow (Suggestion: "Extraversion and introversion are linked to locus of control. Introverts, often marginalized in a society that values social interaction, may achieve better mental health by avoiding unnecessary social engagements".). Moreover, further insights on how extraversion and introversion influence the practice of selective sociality could be provided.
Authors’ response: Thank you for your valuable suggestion. We have added the following sentences to elaborate more on the distinct behaviors of extroverts and introverts:
Extraversion and introversion are deeply connected with locus of control. Extraversion is often associated with external locus of control, whereby introversion is linked with internal locus of control. These connections play a role in shaping how people approach and manage social interactions. Extroverts may thrive on external social engagements. They are typically more outgoing, sociable, and energized by interactions with others. They may be more inclined to perceive that external factor, such as social approval or relationships, significantly influence their success and well-being. This perception can lead to a greater emphasis on social engagements as a means of achieving desired outcomes. Introverts, on the other hand, may achieve better mental health by limiting social interactions to those that are necessary or meaningful. They tend to be more reserved, introspective, and find energy in solitary activities or smaller, more intimate social settings. They are more likely to believe that their personal actions, thoughts, and decisions are the primary determinants of their life experiences. This belief aligns with a preference for self-directed activities and a tendency to limit social engagements to those that are meaningful or necessary.
Conclusions:
I find the conclusions somewhat lengthy, especially their first part. I suggest reconsidering.
Authors’ response: Thank you for your valuable suggestion. Yes, you are right. The initial part of the limitation was too lengthy. We have reduced it now. The modified conclusion is as under:
The current study was mainly focused on developing and validating a new scale on selective sociality. Due to its limited scope, it was unable to infer the causalities of selective sociality. Since the current study was performed within a specific collectivistic culture, its generalizability to other cultures, especially the individualistic Western cultures, is also limited. Moreover, the inevitable individual differences in each culture may also limit the generalizability of our scale. The difference in the values of individualistic and collectivistic cultures would have been addressed more appropriately if data were collected cross-culturally. Future researchers are advised to assess selective sociality in a cross-cultural perspective to analyze item response patterns and social desirability.
General comments:
While the article is well-written and maintains a smooth narrative flow, several elements could benefit from being shortened or simplified to enhance focus (examples were provided above). Additionally, incorporating more examples and references would strengthen the argumentation and provide more clarity and understanding to the readers. By addressing these aspects, the paper could achieve a more significant impact.
Authors’ response: Thank you so much for your positive feedback and valuable input. We took all your comments seriously and carefully addressed all the identified corrections in the manuscript. We have thoroughly reviewed and revised the relevant sections to ensure accuracy and clarity. By incorporating your valuable feedback, we believe that the quality and effectiveness of our study have been significantly improved. We sincerely appreciate your recognition of the novelty and interest of our research. Your guidance and input have been invaluable in refining our work, and we are grateful for your support. Please be assured that we have addressed all your comments and suggestions in their true spirit.

Round 2
Reviewer 1 Report
Comments and Suggestions for Authors
The revised version is a marked improvement in relation to the pervious one. However I would like to see a few revisions to which I pointed out earlier but were not addressed by the authors:
1. Check the manuscript for grammatical issues - they are minor but they are there (e.g. in the abstract "Selective sociality is a crucial psychosocial skill involving the intentional 23 pursuit that involves intentionally seeking out or avoid specific individuals, places, or 24 activities").
2. I still would like to see more detail about the origins of the measure's items - where were they taken from? who phrased them? were they checked for content validity?
3. I still disagree with the authors examination of divergent validity -and leave it to the editor to make a decision on this.
Other than that, I believe that when these issues will be addressed - the manuscript may be published.
Comments on the Quality of English LanguageSee my comment above.
Author Response
Reviewer 1
The revised version is a marked improvement in relation to the pervious one. However I would like to see a few revisions to which I pointed out earlier but were not addressed by the authors:
Authors’ response: Thank you again for your positive feedback and valuable input. We assure you that we took all your comments very seriously. The points which you mention in the second revision have been rechecked by us and modified in accordance with your suggestions. Your guidance and input have been invaluable in refining our work, and we are grateful for your support.
- Check the manuscript for grammatical issues - they are minor but they are there (e.g. in the abstract "Selective sociality is a crucial psychosocial skill involving the intentional 23 pursuit that involves intentionally seeking out or avoid specific individuals, places, or 24 activities").
Authors’ response: Thank you again for your feedback. The sentence you mentioned in the abstract is modified as: Selective sociality is a psychosocial skill to approach or avoid a person, situation, or content with a genuine intention to enhance psychosocial health. We have rechecked all grammatical issues in the paper and have modified those in several places.
- I still would like to see more detail about the origins of the measure's items - where were they taken from? who phrased them? were they checked for content validity?
Authors’ response: Thank you for your concerns. The text under heading Selective Sociality Scale has been improved. We added the details about item-construction and measuring their face validity. The modified text is as under:
The SSS was developed and validated in the current series of four consecutive studies. The initial item pool for the SSS consisted of 24 items that were constructed on based on a person’s interaction with the public, friends, and family. The items developed for measuring selective sociality were constructed through a systematic and theory-driven process, grounded in the conceptual framework of selective sociality. This concept emphasizes the intentional selection of social interactions to optimize psychosocial health. Each item reflects specific behaviors or attitudes associated with selective sociality, such as the preference for a limited social circle, prioritization of family over friends, and tendencies towards social engagement or withdrawal. The items were designed to capture the multidimensional nature of selective sociality, including both the quantity and quality of social interactions. The process began with an extensive literature review on selective sociality and related constructs, such as social networks, social engagement, and psycho-social health. This review informed the initial pool of items, ensuring they were reflective of key theoretical aspects. Moreover, the items were crafted to balance positive and negative social behaviors, allowing for a comprehensive assessment of selective sociality. For example, items like "I have a very limited circle of friends" and "I have a very vast circle of friends" capture opposing ends of the sociality spectrum, while items like "I focus more on my family than my friends or colleagues" and "I focus more on my friends or colleagues than my family" explore different social priorities. These items were observed by a panel of five PhD holder researchers with strong expertise in psychology, statistics and psychometrics, who determined the items to have appropriate face validity for the construct of selective sociality.
- I still disagree with the authors examination of divergent validity -and leave it to the editor to make a decision on this.
Authors’ response: We respect your valuable disagreement. To avoid this confusion, we have now simply used the term “correlation” instead of divergent validity. Several modifications have been made in this regard throughout the paper. You will not find the words “convergent” or “discriminant” or “divergent” in the revised paper. Significant positive and inverse correlations have been used to discuss this. The modifications done in this regard are reflected at lines 29-38; 267-270; 338-342; 344; 351; 358; 517-538.
Other than that, I believe that when these issues will be addressed - the manuscript may be published.
Authors’ response: We thank you for your valuable feedback which really improved our paper. Once again, we assure you that we have addressed all your comments in the revised manuscript.

Reviewer 3 Report
Comments and Suggestions for Authors
Dear Authors,
I have read the amended paper carefully. I appreciate the effort and professionalism you have put into this work. Thank you for your dedication. I will now share my observations with you:
The new subject of the paper is "Avoid and Rule: The Development and Validation of the Selective Sociality Scale" (thank you for this modification!). Although the Introduction has been significantly developed (e.g., lines 44 – 77, 206 - 256), the provided content is irrelevant to the discussed topic (e.g., section "1.3. Recent changes in the psychosocial environment through information technology" could be significantly shortened or removed). Consequently, the Introduction's function, which describes the studied construct, has not been achieved. This lack of specificity could be misleading to potential readers. Moreover, the Introduction is lengthy (almost 5.5 pages), negatively affecting the paper's focus. I suggest reconsidering and providing a straightforward, highly relevant, straight-to-the-point (related to the studied construct) narrative.
Moreover, the Selective Sociality Scale per sé and the Scoring Instructions provided in the Supplementary Materials demonstrate several inconsistencies. First, it is unclear how the items apply specifically to the presented construct because of their vagueness (e.g., "I usually talk too much", "I have a very vast circle of friends"). Therefore, they could be used to assess, e.g. extraversion. Furthermore, given the lack of specificity, the items demonstrate a general tendency rather than a specific dimension. Second, the items assigned to the sub-scales (Public, Friends, Family) do not align clearly with their respective domains. The following items: "I do not like talking a lot", "I usually talk too much", and "I like talking too much", may reflect general social engagement (subscale: "Public") while are included in the different subscales. Finally, the items are very similar within the separate dimensions (e.g., subscale Public: "I do not like talking a lot", "I have a very limited circle of friends", "I focus more on my family than my friends or colleagues" vs subscale: "I usually talk too much", "I like too many friendships", "I spend most of my time with my close family").
Ultimately, your best effort to improve the Conclusions section is not unnoticed. However, once again, the first part of this section (lines 665 – 670) is irrelevant to the discussed topic. Please reinforce the key messages in order for the readers to take away the most important information. Currently, the conclusions do not sufficiently highlight the core findings and implications of the study, which could leave readers unclear about the significance of the Selective Sociality Scale.
Kind regards
Author Response
Reviewer 3
Dear Authors,
I have read the amended paper carefully. I appreciate the effort and professionalism you have put into this work. Thank you for your dedication. I will now share my observations with you:
Authors’ response: Thank you again for your positive feedback and valuable input. We assure you that we took all your comments very seriously. The points which you mention in the second revision have been rechecked by us and modified in accordance with your suggestions. Your guidance and input have been invaluable in refining our work, and we are grateful for your support.
The new subject of the paper is "Avoid and Rule: The Development and Validation of the Selective Sociality Scale" (thank you for this modification!). Although the Introduction has been significantly developed (e.g., lines 44 – 77, 206 - 256), the provided content is irrelevant to the discussed topic (e.g., section "1.3. Recent changes in the psychosocial environment through information technology" could be significantly shortened or removed). Consequently, the Introduction's function, which describes the studied construct, has not been achieved. This lack of specificity could be misleading to potential readers. Moreover, the Introduction is lengthy (almost 5.5 pages), negatively affecting the paper's focus. I suggest reconsidering and providing a straightforward, highly relevant, straight-to-the-point (related to the studied construct) narrative.
Authors’ response: The title of the paper was modified to address your valuable suggestions. Based on the suggestions we received from you and two other reviewers, we modified the Introduction significantly. We rearranged the sequence of the themes, inserted heading and sub-headings, and added almost 60 more citations. Please note that “selective sociality” is a new construct and based on the suggestions of other reviewers, we explained the theoretical development of this construct i.e. how we reached to this construct exploring the earlier theories, models, and paradigms on mental health. In the Introduction, our purpose is to provide the readers with a brief hence comprehensive background on the traditional and contemporary views on mental health, along with the rapid advancements in information technology; so that we could establish the utility and usefulness of selective sociality. We request you to please accommodate this theoretical foundation as the journal has no restrictions on the length. We hope you will sense our intentions by accommodating this. We thank you for your understanding.
Moreover, the Selective Sociality Scale per sé and the Scoring Instructions provided in the Supplementary Materials demonstrate several inconsistencies. First, it is unclear how the items apply specifically to the presented construct because of their vagueness (e.g., "I usually talk too much", "I have a very vast circle of friends"). Therefore, they could be used to assess, e.g. extraversion. Furthermore, given the lack of specificity, the items demonstrate a general tendency rather than a specific dimension. Second, the items assigned to the sub-scales (Public, Friends, Family) do not align clearly with their respective domains. The following items: "I do not like talking a lot", "I usually talk too much", and "I like talking too much", may reflect general social engagement (subscale: "Public") while are included in the different subscales. Finally, the items are very similar within the separate dimensions (e.g., subscale Public: "I do not like talking a lot", "I have a very limited circle of friends", "I focus more on my family than my friends or colleagues" vs subscale: "I usually talk too much", "I like too many friendships", "I spend most of my time with my close family").
Authors’ response: Thank you for analyzing and commenting on this. We have analyzed the content of your query and comments and have concluded that the numbering of the items (different in tables and in the supplementary file) may be the cause of confusion in this regard.
Please note that, based on another reviewer’s comments, we have not added details on the item-construction as follow:
The SSS was developed and validated in the current series of four consecutive studies. The initial item pool for the SSS consisted of 24 items that were constructed on based on a person’s interaction with the public, friends, and family. The items developed for measuring selective sociality were constructed through a systematic and theory-driven process, grounded in the conceptual framework of selective sociality. This concept emphasizes the intentional selection of social interactions to optimize psychosocial health. Each item reflects specific behaviors or attitudes associated with selective sociality, such as the preference for a limited social circle, prioritization of family over friends, and tendencies towards social engagement or withdrawal. The items were designed to capture the multidimensional nature of selective sociality, including both the quantity and quality of social interactions. The process began with an extensive literature review on selective sociality and related constructs, such as social networks, social engagement, and psycho-social health. This review informed the initial pool of items, ensuring they were reflective of key theoretical aspects. Moreover, the items were crafted to balance positive and negative social behaviors, allowing for a comprehensive assessment of selective sociality. For example, items like "I have a very limited circle of friends" and "I have a very vast circle of friends" capture opposing ends of the sociality spectrum, while items like "I focus more on my family than my friends or colleagues" and "I focus more on my friends or colleagues than my family" explore different social priorities. These items were observed by a panel of five PhD holder researchers with strong expertise in psychology, statistics and psychometrics, who determined the items to have appropriate face validity for the construct of selective sociality.
Moreover, please note that the structure of the sub-scales (public, friends, and family) is as follows:
Item No. |
Item |
|
Public |
|
|
SSS4 |
I usually talk too much.* |
|
SSS7 |
I like talking too much.* |
|
SSS1 |
I don’t like talking a lot. |
|
Friends |
|
|
SSS3 |
I focus more on my family than my friends or colleagues. |
|
SSS6 |
I spend most of my time with my close family. |
|
SSS9 |
I focus more on my friends or colleagues than my family.* |
|
Family |
|
|
SSS8 |
I have a very vast circle of friends.* |
|
SSS2 |
I have a very limited circle of friends. |
|
SSS5 |
I like too many friendships.* |
|
* reversed coded
Yes, you are right that items 1, 4, 7 are general. These refer to social interactions with general public. However, items 3, 6, 9 specifically focus on friends; and items 2, 5, 8 specifically focus on family. Extraversion is a similar construct. However, the items of the SSS are related to selective sociality. Significant inverse correlations between the SSS and the Extraversion (sub-scale of the Big-Five Inventory) and the External Locus of Control further establish that the construct of selective sociality (the combined 9 items) is statistically distinct from extraversion and external locus of control. We believe that, by typo mistake is the supplementary file, this confusion might have occurred. We hope that we are now able to clarify this and satisfy your queries. The tables and the figures further explain this.
Ultimately, your best effort to improve the Conclusions section is not unnoticed. However, once again, the first part of this section (lines 665 – 670) is irrelevant to the discussed topic. Please reinforce the key messages in order for the readers to take away the most important information. Currently, the conclusions do not sufficiently highlight the core findings and implications of the study, which could leave readers unclear about the significance of the Selective Sociality Scale.
Authors’ response: We realized this and removed the unnecessary initial sentences from the conclusion. The modified conclusion is as follows:
The current paper reflects the findings of four consecutive studies that developed and validated the selective sociality scale. Selective sociality is a psychosocial skill to approach or avoid a person, situation, or content with a genuine intention to enhance psychosocial health. The SSS consists of three subscales based on an individual’s interaction with the public, friends, and family. These studies establish the reliability and validity of the SSS. Understanding and practicing selective sociality requires a comprehensive understanding of conformity, compliance with social norms, locus of control, extraversion, introversion, avoidance, and escape. The paper concludes with a call for education on these matters by mental health professionals, influencers, and policymakers.
Kind regards
Authors’ response: Once again, thank you for taking time and evaluating the paper. Your valuable feedback has helped us a lot in improving our paper.

Round 3
Reviewer 3 Report
Comments and Suggestions for Authors
Thank you very much for your extensive replies.
The authors mentioned that the comments have been considered and amendments have been made accordingly, however, the paper has not been changed significantly. For instance, the introduction is still irrelevant and very long (6 pages).
The development process of the scale has flaws regarding its content validity. The items of the scale are not specific, and the measured construct seems to not identifiable by this scale.
Author Response
Thank you very much for your extensive replies.
Authors’ response: Thank you again for your positive feedback and valuable input. We assure you that we took all your comments very seriously. The points which you mention in the second revision have been rechecked by us and modified in accordance with your suggestions. Your guidance and input have been invaluable in refining our work, and we are grateful for your support.
The authors mentioned that the comments have been considered and amendments have been made accordingly, however, the paper has not been changed significantly. For instance, the introduction is still irrelevant and very long (6 pages).
Authors’ response: We appreciate your concerns. Since in review 1 and 2, one of the reviewers believed we give a detailed background on how we reached the construct of selective sociality. Therefore, we provided detailed information. However, based on your extensive review, we have now reduced the introduction to a significant degree; whereby we reduced the sub-headings and text. We have now kept the most relevant text. Referencing has also been drastically reduced. Our modified Introduction is of 1362 words (from the earlier 2554 words). Citations are 98 (from the earlier 163). We are quite optimistic that you will agree to this reduction.
The development process of the scale has flaws regarding its content validity. The items of the scale are not specific, and the measured construct seems to not identifiable by this scale.
Authors’ response: We are really grateful to you for emphasizing again and again on this issue. We accept the mistakes in the previous versions of the paper. There were two major mistakes that caused all confusion in this regard. Firstly, the numbering of the items had typing errors. This was due to the sequence of item numbering of the original scale and the item numbering of the EFA and CFA. The item numbers in the original scale (as provided in the supplementary file) were not based on the factor structure. However, in the tables we placed items based on their factors to keep items of the same sub-scale together. Secondly the labeling of the sub-scales also had been mistaken whereby friends and family were mistakenly mixed with each other, only in the tables 2 and 3. In the figures, these were correct. The tables and figures, therefore, were not aligned with each other. We have revised all the relevant instances in this regard i.e. within text and tables. The item numbering has also been rearranged in tables 2 and 3 in numeric sequence, while keeping the items of one sub-scale together.
Table 2: Exploratory factor analysis (Study 1; n = 221)
Item No. |
Item |
Factor Structure |
Extraction |
Item-total correlation |
Item-scale correlation |
||||
Public |
Friends |
Family |
Public |
Friends |
Family |
||||
SSS1 |
I don’t like talking a lot. |
0.724 |
0.486 |
0.214 |
0.526 |
.670** |
.858** |
.475** |
.190** |
SSS4 |
I usually talk too much.* |
0.852 |
0.535 |
0.272 |
0.727 |
.727** |
.901** |
.508** |
.255** |
SSS7 |
I like talking too much.* |
0.918 |
0.638 |
0.311 |
0.845 |
.787** |
.908** |
.605** |
.290** |
SSS2 |
I have a very limited circle of friends. |
0.545 |
0.786 |
0.337 |
0.62 |
.745** |
.519** |
.879** |
.317** |
SSS5 |
I like too many friendships.* |
0.574 |
0.748 |
0.39 |
0.573 |
.750** |
.540** |
.837** |
.358** |
SSS8 |
I have a very vast circle of friends.* |
0.527 |
0.864 |
0.413 |
0.751 |
.762** |
.490** |
.891** |
.386** |
SSS3 |
I focus more on my family than my friends or colleagues. |
0.31 |
0.411 |
0.869 |
0.756 |
.624** |
.288** |
.378** |
.893** |
SSS6 |
I spend most of my time with my close family. |
0.113 |
0.249 |
0.725 |
0.545 |
.447** |
0.095 |
.220** |
.843** |
SSS9 |
I focus more on my friends or colleagues than my family.* |
0.371 |
0.491 |
0.807 |
0.669 |
.672** |
.333** |
.458** |
.871** |
Table 3: Confirmatory factor analysis (Study 2; n = 135)
Factor |
Item |
Factor loadings |
|
Residual variances |
|||||||||||
Estimate |
SE |
z |
p |
|
Estimate |
SE |
z |
p |
|||||||
Public |
SSS1 |
0.740 |
0.139 |
9.510 |
< 0.001 |
|
0.452 |
0.214 |
6.727 |
< 0.001 |
|||||
|
SSS4 |
0.869 |
0.132 |
11.769 |
< 0.001 |
|
0.245 |
0.186 |
4.181 |
< 0.001 |
|||||
|
SSS7 |
0.833 |
0.125 |
11.072 |
< 0.001 |
|
0.306 |
0.169 |
5.039 |
< 0.001 |
|||||
Family |
SSS2 |
0.790 |
0.116 |
10.556 |
< 0.001 |
|
0.376 |
0.139 |
6.512 |
< 0.001 |
|||||
|
SSS5 |
0.801 |
0.119 |
10.687 |
< 0.001 |
|
0.358 |
0.147 |
6.122 |
< 0.001 |
|||||
|
SSS8 |
0.904 |
0.114 |
12.776 |
< 0.001 |
|
0.182 |
0.130 |
3.592 |
< 0.001 |
|||||
Friends |
SSS3 |
0.902 |
0.108 |
11.851 |
< 0.001 |
|
0.187 |
0.144 |
2.626 |
0.009 |
|||||
|
SSS6 |
0.743 |
0.113 |
9.465 |
< 0.001 |
|
0.448 |
0.143 |
6.519 |
< 0.001 |
|||||
|
SSS9 |
0.737 |
0.104 |
9.045 |
< 0.001 |
|
0.457 |
0.124 |
6.006 |
< 0.001 |
|||||
Further regarding content validity, we have also modified the definition of selective sociality and have removed situations and content from the definition. The new definition of selective sociality, everywhere in the revised manuscript is: “Selective sociality is a psychosocial skill that emphasizes consciously engaging and speaking more or less with selected individuals.”. This new definition is highly aligned with the items of the scale.
We have now provided more details on the items and linked them with the construct of selective sociality. The new sentences added in this regard are as follows:
The nine items of the SSS are categorized by the frequency of communication and the relational nature of social interaction. Item 1 (I don’t like talking a lot), item 4 (I usually talk too much; reversely coded), and item 7 (I like talking too much; reversely coded) are related to communication in general. These items are not specific to family and friends. They project if the person talks too much or a little to evaluate social interaction through general communication. Through these items, we assumed that the socially selective person would talk lesser. Therefore, a higher score on these items would reflect higher selective sociality in general social interactions. Item 3 (I focus more on my family than my friends or colleagues), item 6 (I spend most of my time with my close family), and item 9 (I focus more on my friends or colleagues than my family; reversely coded) are related to one’s interaction with family. Through these items, we assumed that the socially selective person would spend more time with their family as compared to friends. Therefore, a higher score on these items would reflect higher selective sociality, while focusing more on family than friends. Item 2 (I have a very limited circle of friends), item 5 (I like too many friendships; reversely coded), and item 8 (I have a very vast circle of friends; reversely coded) are related to one’s interaction with friends. Through these items, we assumed that the socially selective person would spend less time with their friends. Therefore, a higher score on these items would reflect higher selective sociality in friendships. In conclusion, an overall higher score on all the nine items of the SSS would reflect higher selective sociality, as evaluated through the respondent’s communication and interaction with family and friends.
We hope that these corrections, which are really needed and were performed only due to your keen vigilance, will solve the confusion. The harmony between the definition of selective sociality and the items of the SSS has been achieved.
We are thankful to you for pointing out these mistakes. Their correction was surely needed. We are quite optimistic that you will accept these changes with positive recommendation.
